# Cross-talk between microglia and neurons regulates HIV latency

**David Alvarez-Carbonell**[1], **Fengchun Ye**[1], **Nirmala Ramanath**[1], **Yoelvis Garcia-Mesa**[1¤], **Pamela E. Knapp**[2], **Kurt F. Hauser**[2], **Jonathan Karn**[1]*

**1** Department of Molecular Biology and Microbiology, Case Western Reserve University, Cleveland, Ohio, United States of America, **2** Departments of Pharmacology and Toxicology and Anatomy and Neurobiology, Virginia Commonwealth University School of Medicine, Richmond, Virginia, United States of America

¤ Current address: Department of Neurology, University of Pennsylvania, Philadelphia, Pennsylvania, United States of America

* jonathan.karn@case.edu

**Data Availability Statement:** All relevant data are within the manuscript and its Supporting Information files.

## Abstract

Despite effective antiretroviral therapy (ART), HIV-associated neurocognitive disorders (HAND) are found in nearly one-third of patients. Using a cellular co-culture system including neurons and human microglia infected with HIV (hμglia/HIV), we investigated the hypothesis that HIV-dependent neurological degeneration results from the periodic emergence of HIV from latency within microglial cells in response to neuronal damage or inflammatory signals. When a clonal hμglia/HIV population (HC69) expressing HIV, or HIV infected human primary and iPSC-derived microglial cells, were cultured for a short-term (24 h) with healthy neurons, HIV was silenced. The neuron-dependent induction of latency in HC69 cells was recapitulated using induced pluripotent stem cell (iPSC)-derived GABAergic cortical (iCort) and dopaminergic (iDopaNer), but not motor (iMotorNer), neurons. By contrast, damaged neurons induce HIV expression in latently infected microglial cells. After 48–72 h co-culture, low levels of HIV expression appear to damage neurons, which further enhances HIV expression. There was a marked reduction in intact dendrites staining for microtubule associated protein 2 (MAP2) in the neurons exposed to HIV-expressing microglial cells, indicating extensive dendritic pruning. To model neurotoxicity induced by methamphetamine (METH), we treated cells with nM levels of METH and suboptimal levels of poly (I:C), a TLR3 agonist that mimics the effects of the circulating bacterial rRNA found in HIV infected patients. This combination of agents potently induced HIV expression, with the METH effect mediated by the σ1 receptor (σ1R). In co-cultures of HC69 cells with iCort neurons, the combination of METH and poly(I:C) induced HIV expression and dendritic damage beyond levels seen using either agent alone, Thus, our results demonstrate that the cross-talk between healthy neurons and microglia modulates HIV expression, while HIV expression impairs this intrinsic molecular mechanism resulting in the excessive and uncontrolled stimulation of microglia-mediated neurotoxicity.

**Funding:** This work was supported by NIH grants, R01 DA036171, R01 DA043159, R01 DA049481 (JK PK KFH), R01 MH113457 (Paula Cannon and JK) and P30 AI36219, CWRU/UH Center for AIDS Research (JK). The funders had no role in study design, data collection and analysis, decision to publish, or preparation of the manuscript.

**Competing interests:** The authors have declared that no competing interests exist.

## Author summary

Now that HIV patients are living longer due to improved anti-retroviral therapy, the prevalence of HIV-associated neurocognitive disorders (HAND) is rising. HAND is linked to chronic inflammation and microgliosis and is exacerbated by substances of abuse such as methamphetamine (METH). Using co-culture models of neurons and microglia, we demonstrate that healthy neurons can suppress HIV transcription in infected microglia over a short-term (24 h). By contrast, damaged neurons reactivate latent HIV expression. Thus, over a longer term (72 h), neurons lose the ability to suppress HIV expression, resulting in enhanced neural injury. This damaging cycle of latency reversal and neuronal degeneration can be exacerbated by METH and inflammation. Our results support the hypothesis that the dendritic simplification and neuronal injury associated with HAND results from viral reactivation induced by neuronal and microglial cross-talk and exogenous inflammatory stimuli.

## Introduction

Approximately 30% of the HIV-infected individuals on combination anti-retroviral treatments (cART) display symptoms of cognitive impairment and central nervous system (CNS) pathology, a syndrome known as HIV-associated neurocognitive disorders (HAND) [1–5]. While the incidence of outright HIV-associated dementia (HAD), which is due to the neurological damage induced by actively replicating HIV, has sharply declined due to effective cART, the prevalence of the milder HAND conditions remains high [5, 6], and is expected to increase further as the HIV infected population ages [7]. In the US, 10–15% of HIV patients acknowledge METH use [8], which exacerbates the effects of HIV infection in the CNS [9, 10] due to a combination of neurotoxic effects and the enhancement of HIV replication in microglia [11–14]. There is compelling evidence that HIV exacerbates age-associated cognitive decline and diminishes neuropsychological performance across multiple cognitive domains [7, 15].

HIV-1 replication in the CNS is initiated from invading monocytes and CD4$^+$ T cells, and then spreads to microglial cells and, arguably, astrocytes within the brain parenchyma [16–27]. Definitive evidence that HIV replicates in myeloid lineage cells within the CNS comes from the observation that HAD patients harbor macrophage-tropic HIV-1 variants that grow selectively in the CNS [21, 28–31]. A consequence of HIV-1 replication in longer-lived cell types in the brain, including microglia, is that virus is depleted more slowly in the cerebrospinal fluid (CSF) than virus in the peripheral circulation after the initiation of therapy [32]. Minimal viral replication still persists in the CNS [33, 34], especially in microglia and perivascular macrophages [35], in part because not all anti-HIV drugs are able to cross the blood-brain barrier with high efficiency. However, latent infections of microglial cells and astrocytes also appear to contribute to the long-term persistence of HIV in the CNS [36–44].

Microglia constitute the first barrier of the innate immune response in the CNS [45]. They constantly survey the brain parenchyma to detect physiological changes and then migrate to regions of damage, where they become activated [46]. Normally, the activation of microglia in response to inflammatory stimuli is characterized by a transition from a resting state (M0 cells) to an activated proinflammatory phenotype (M1 cells). It is generally believed that over-activated microglia exacerbate neuronal injury through the synthesis and secretion of proinflammatory and cytotoxic factors [47, 48]. Therefore, in NeuroHIV, microglia-mediated neuronal injury appears to result from excitotoxicity [49, 50], which disrupts the intrinsic

molecular mechanisms that control ion homeostasis and energy production in neurons [51–53].

HIV-1 neuropathology arises because of the combined neurotoxic effects of the viral proteins and exaggerated inflammatory responses by microglial cells. The HIV proteins Tat [54–67], gp120 [66, 68–71], Vpr [72, 73], and Nef [74–78] can directly induce neuronal damage. Additionally, microglia can contribute to neurodegeneration through the release of cytokines and toxins that damage neurons and astrocytes [79–81]. The natural control mechanisms preventing over-activation of microglial cells are likely to be impaired as a consequence of HIV infection. HIV-infected microglia respond vigorously to proinflammatory signals and produce an excess of cytokines [37, 39]. Importantly, neuronal dysfunction does not correlate with the number of HIV-infected cells or viral antigens in CNS [82, 83], but rather with elevated inflammatory cytokine levels. Elevated TNF-$\alpha$ mRNA levels in microglia and astrocytes [82, 84] and high levels of IL-1$\beta$ and TNF-$\alpha$ are seen in the CNS of patients with HAD [85, 86]. Similarly, increased IL-6 and IL-8 levels are found in the brains of HIV-1 infected patients [4, 87]. Immune-activated HIV-infected, brain-infiltrating macrophages and resident microglia release high levels of neurotoxic cytokines such as TNF-$\alpha$ and IL-1$\beta$ [88].

Co-culture systems have been used extensively to study neuron-glia interactions and, because of the ability to measure the contribution of each cell type individually, are highly valuable tools to understand neuronal-microglial cross-talk [89–96]. Here, we developed a series of co-culture systems to study the impact of HIV expression in infected microglial cells on their interactions with neurons. Using LUHMES-derived neurons, which have been extensively use to study neurodegenerative processes because of their dopaminergic-like features [97], and iPSC-derived primary neurons, in co-culture with hµglia/HIV as a platform to study the effect of neuronal and microglial interactions on the regulation of HIV expression in the CNS, we found unexpectedly that healthy neurons induce HIV silencing in microglia and prevent spontaneous HIV reactivation in latent hµglia/HIV cells. By contrast, damaged neurons increased HIV expression in microglial cells. In addition, we found that METH, which can induce HIV expression in hµglia/HIV cells in a $\sigma$1R-dependent manner, sensitizes microglial cells to proinflammatory agents and exacerbates neurodegeneration in the neuronal-microglial co-cultures. We conclude that HIV expression disrupts the normal interplay between microglia and neurons and thereby induces neurodegeneration.

## Results

### Establishment of a co-culture system between LUHMES-derived neurons and µglia/HIV cells

We have shown previously that HIV readily establishes latency in immortalized human microglial cells [36–39]. In order to understand how neuronal and microglial cross-talk regulates HIV latency in the brain, we first established a co-culture system between LUHMES-derived neurons and human microglia infected with HIV reporter viruses (**Fig 1**). Undifferentiated Lund human mesencephalic (LUHMES) cells, obtained from ATCC (CRL-2927) and originally developed and characterized at Lund University (Lund, Sweden) [98, 99], carrying a vector expressing red fluorescent protein (RFP) to allow cell visualization were kindly provided by Dr. Stefan Schildknecht (Konstanz, Germany) [100]. The cells were expanded in neuronal growth medium (NGM) for 4 days. After 4-days, NGM was replaced by a modified neuronal differentiation medium (mNDM), to inhibit the expansion of neuronal precursors and induce differentiation into dopaminergic neurons [99]. After differentiation for 1 day, immortalized human microglial cells (hµglia) infected with an HIV vector carrying a green fluorescence protein (GFP) reporter (e.g. HC69 cells) [36, 37, 39], were plated on top of the neurons, typically

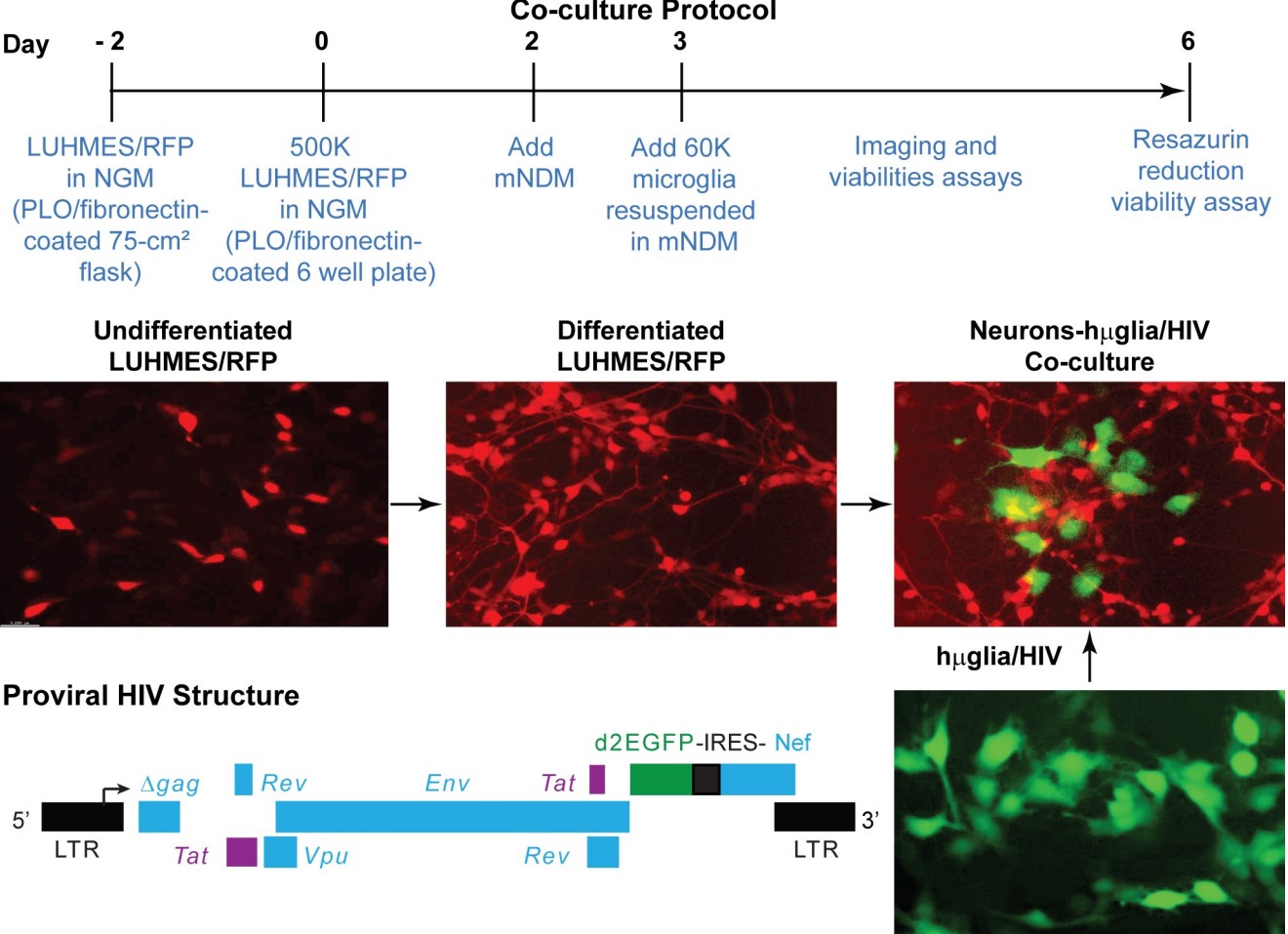

**Fig 1. Establishment of a neurons-hμglia/HIV co-culture system.** The time line is indicated by the large black arrow, with the corresponding day number shown in red. Undifferentiated, neuron precursor cells LUHMES/RFP are plated and allowed to expand for 2 days prior to transferring to the experimental wells at 500,000 neurons per well of a six-well plate at day 0. After 2 days, the neuronal differentiation is initiated by adding modified neuronal differentiation medium (mNDM). After differentiation for 1 day, 60,000 immortalized human microglial cells (hμglia/HIV) carrying a single round HIV construct with a GFP reporter (diagrammed, bottom left) are added. The viability of the neurons and the expression levels of HIV are monitored for 3 days.

at a ratio of 3 microglia per 25 neurons (**Fig 1**). The co-cultures could be maintained for up to 96 h and HIV expression and neuronal damage quantitatively and qualitatively evaluated using molecular, biochemical, flow cytometric, and morphologic endpoints.

A critical technical step in developing the co-culture system was to develop a culture medium that maximized survival for both the neurons and the microglia (**S1 Fig**). We found that a low concentration of fetal bovine serum (FBS 0.2%), was optimal to maintain microglial viability, but higher concentrations of FBS were toxic for the neurons. To further preserve microglial viability, the FBS was enriched with 1X insulin-transferrin-sodium selenite supplement in the final mNDM formulation.

## Neurons silence HIV in a density-dependent manner

We have previously shown that HC69 cells undergo spontaneous HIV reactivation due to autocrine expression of TNF-α [39]. HIV reactivation could be potently repressed by addition of the glucocorticoid receptor agonist dexamethasone (DEXA), which blocked both HIV and

proinflammatory cytokine production [39]. To determine the effect of neurons on spontaneous HIV expression, a mixed population of HC69 (hμglia/HIV cells) containing a combination of both GFP$^+$ and GFP$^-$ cells, were co-cultured with LUHMES-derived neurons, as described above. Cultures were established at three different neuronal densities: 100 neurons per mm$^2$, (a ratio 10:6 of neurons:HC69 microglial cells), 300 neurons per mm$^2$ (25:6, neurons:HC69), and 600 neurons per mm$^2$ (50:6, neurons:HC69). HIV expression was then evaluated after 24 h, using microglia cultured without neurons as a control. Flow cytometry and fluorescence microscopy (**Fig 2A**) analyses demonstrated that HIV expression declines as the ratio of neurons:microglia is increased. The impact of neurons on HIV expression was evaluated over a time course ranging from 0 to 36 h (**Fig 2B**). At 36 h, the percent of HIV expression (Y-axis) was reduced significantly from ~45% in HC69 cells cultured without neurons (red line and squares) to ~25% when 100 neurons per mm$^2$ were present (a 10:6 ratio). At higher ratios there was a further reduction to ~8%. There was no significant neurotoxicity in these co-cultures, with neuron viability remaining above ~90%, as measured by the resazurin assay (**Fig 2C**).

The gating strategy used to measure GFP expression in the microglial cells (CD14$^+$ cells) by flow cytometry, and avoid counting neurons, is shown in **S3 Fig**. Gating on hμglia in the FSC vs. SCC scatter plot resulted in ~100% CD14$^+$ cells and can be used instead of staining for CD14.

In control experiments, to rule-out the possibility that the observed decreased in GFP expression in HC69 cells exposed to neurons was due to microglia-specific cell death or toxicity, we measured both the growth rate and the level of toxicity of the microglial population in the presence and absence of neurons. After 24 h co-culture, there was no statistically significant difference in cell growth or toxicity level (Y-axis) between microglia cultured in the presence of neurons and control cultures at any of the three neuronal densities used (X-axis) (**S2 Fig**).

## iPSC-derived GABAergic cortical and dopaminergic neurons induce HIV silencing

To demonstrate that primary neurons are also capable of silencing HIV, HC69 cells were co-cultured with iPSC-derived GABAergic cortical (iCort), dopaminergic (iDopaNer), or motor (iMotorNer) neurons at a ratio of 50 neurons:6 microglia (**Fig 3**), which was the proportion of LUHMES cells to microglia that caused the greatest suppression of HIV expression. As shown in **Fig 3**, the cortical and dopaminergic neurons, but not motor neurons, attenuated HIV expression in HC69 cells, as shown by both fluorescence microscopy (**Fig 3A**) and flow cytometry (**Fig 3B**). HIV expression in HC69 cells was reduced from 52.34 ± 6.09% to 16.36 ± 3.44% ($p < 10^{-7}$), after co-culture with iPSC-derived cortical neurons. Similarly, dopaminergic neurons decreased HIV expression in HC69 cells to 26.34 ± 8.84% ($p < 10^{-7}$). HIV expression was not significantly inhibited by motor neurons (**Fig 3B**).

To verify the differentiation of the neurons, iPSC-derived neurons were analyzed by immunocytochemical detection using phenotype-specific antibodies against GABAergic, dopaminergic, and cholinergic neuronal markers, respectively, glutamate decarboxylase (GAD65/67), dopaminergic transporter (DAT), and acetylcholinesterase (AchE) (**Fig 3C**). The pan-neuronal marker beta-TUJ [101, 102] (**S4 Fig**), dendritic marker MAP2 [103] (**S5 Fig**), and neuronal receptor marker CXCR3 [104] (**S6 Fig**) were used as positive controls. As expected, the three iPSC cell lineages, as well as LUHMES-derived neurons, stained positive for these proteins. The microglial marker CD11b/c [105–107] was used as a negative control and, as expected, all the neurons were CD11b/c-negative (**S7 Fig**). LUHMES-derived neurons stained positive for

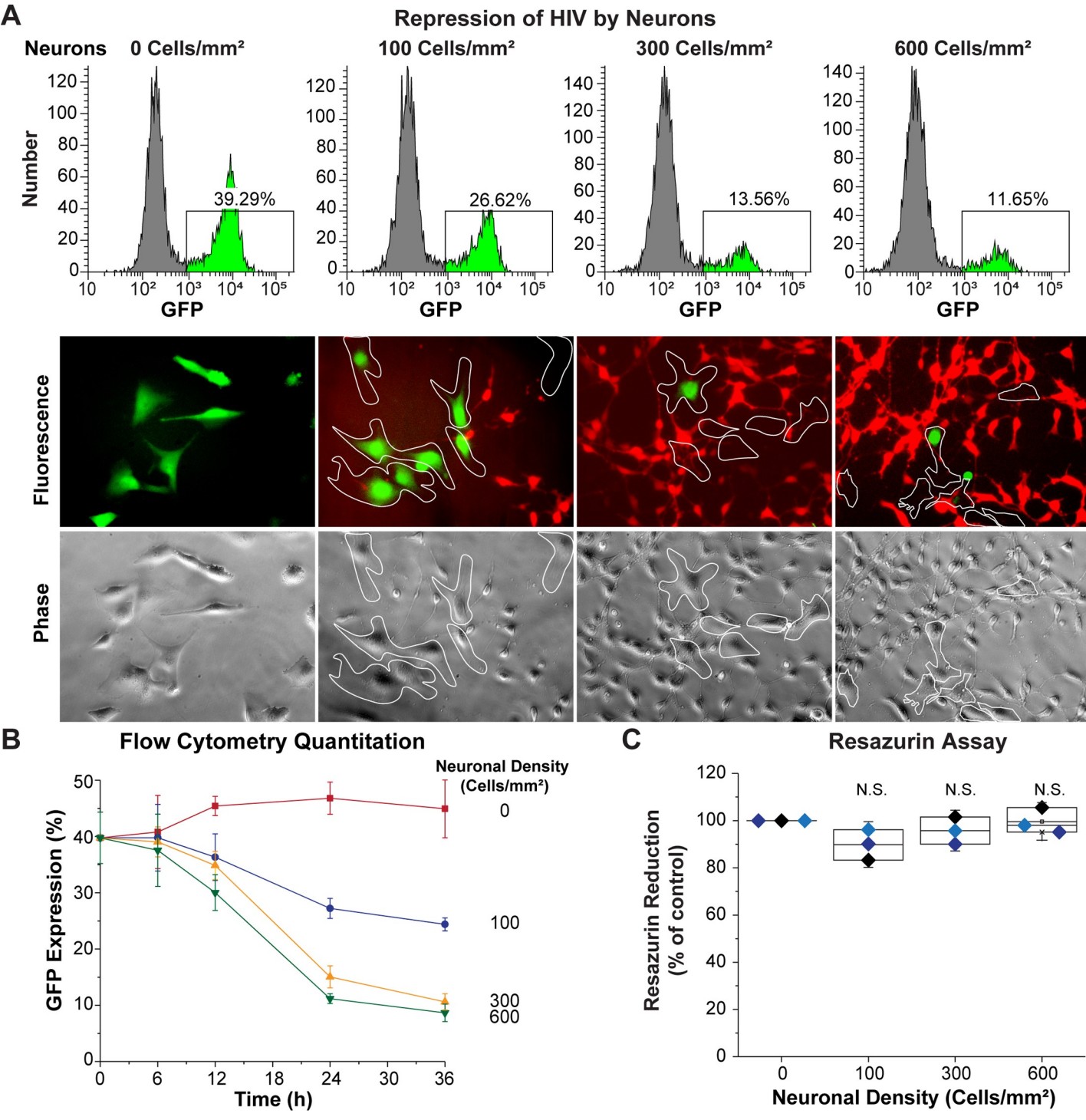

**Fig 2. LUHMES-derived neurons inhibit HIV expression.** 60,000 HC69 (hμglia/HIV) cells were plated in the absence or presence of increasing densities of LUHMES-derived neurons (Cells/mm²). The level of HIV expression was evaluated after 24 h by flow cytometry and fluorescence microscopy. (**A**) Flow cytometry profiles from representative single cultures (top) and microscopy (bottom). In the histograms GFP⁺ cells are indicated in *bright green*. Microglia were identified by phase contrast microscopy and are outlined by the white contours on the micrographs. (**B**) Progressive inhibition of HIV expression (Y-axis) at increasing neuronal densities during a time-course of 36 h (X-axis). The error bars represent the SD of $n$ = 3 independent experiments. (**C**) Resazurin assay to evaluate neuronal viability. The resazurin reduction values (Y-axis) were normalized to the control culture of neurons alone. Each colored symbol represents one experiment. There was no statistically significant (N.S.) differences in cell viability in these experiments.

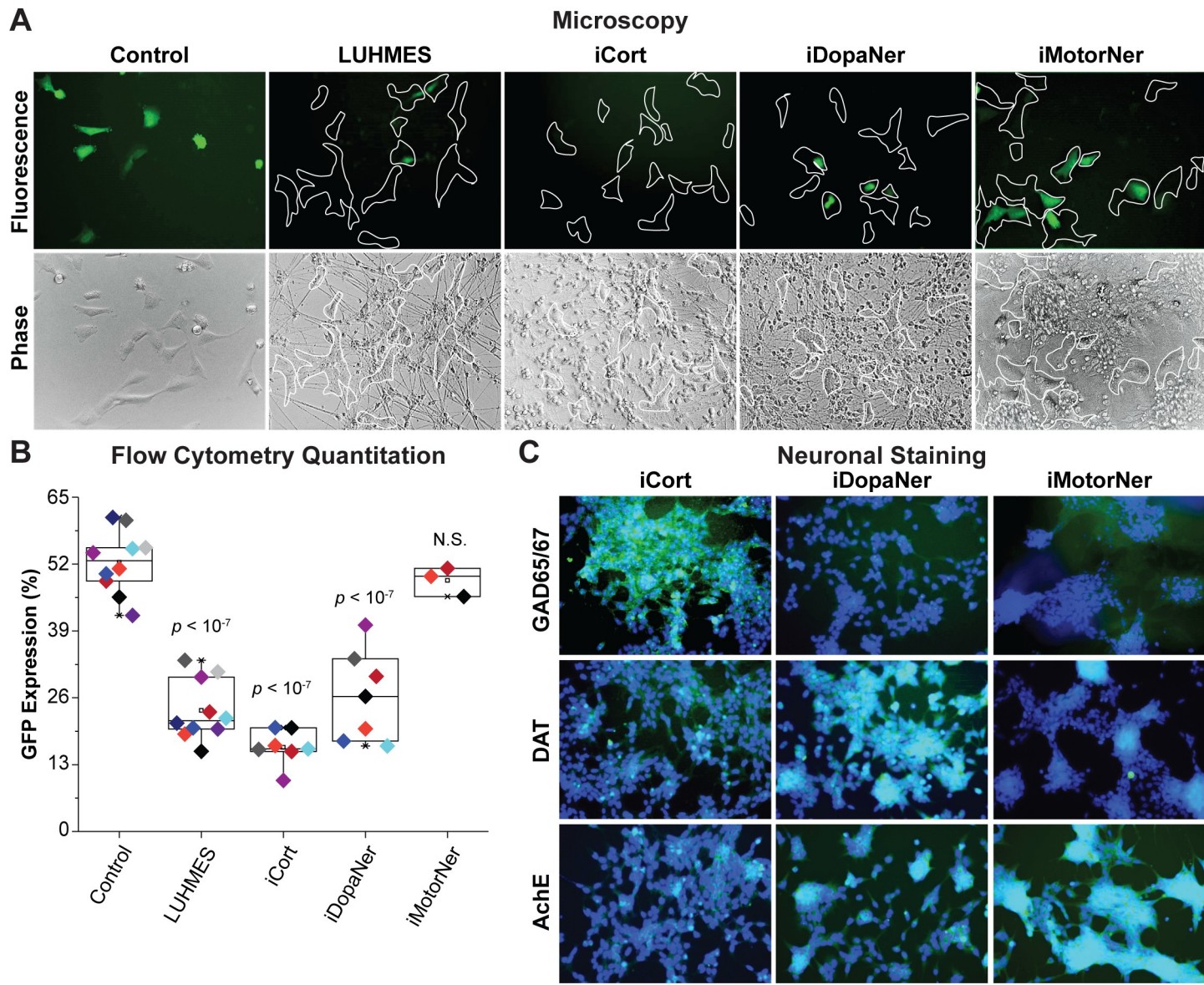

**Fig 3. iPSC-derived neurons repress HIV expression.** (**A**) 60,000 hµglia/HIV HC69 cells were plated in the presence of 0.5 x 10⁶ LUHMES-derived neurons (as positive control) or 0.5 x 10⁶ iPSC-derived GABAergic cortical (iCort), dopaminergic (iDopaNer) or motor neurons (iMotorNer). HIV expression was evaluated after 24 h by fluorescence microscopy. Microglia identified by phase contrast microscopy are outlined by the white contours. (**B**) Flow cytometric analysis of microglial cell GFP expression. The *p*-values of pair-sample, Student's *t*-tests comparing the microglial cells cultured alone or in the presence of neurons are shown. Individual independent experiments are color coded (*n* = number of independent samples). N.S.: non-significant. (**C**) Identification of differentiation of iPSC-derived neurons. Super-imposed images of DAPI stained nuclei (blue) and Alexa-Fluor 488 stained neuronal antigens (green) are shown. Neurons were stained with antibodies against GAD65/67, DAT, and AchE and Alexa Fluor 488-conjugated anti-rabbit secondary antibody.

all three phenotype-specific markers GAD65/67 (**S8 Fig**), DAT (**S9 Fig**), and AchE (**S10 Fig**). iPSC-derived GABAergic cortical neurons strongly expressed GAD65/67, a specific marker of inhibitory GABAergic neurons [108], and only expressed DAT or AchE at low levels (**Fig 3C** and **S8**–**S10 Figs**). iPSC-derived dopaminergic neurons strongly expressed DAT, a dopaminergic neuron-specific marker [109, 110], and had reduced levels of AchE, and very low levels of GAD65/67 (**Fig 3C** and **S8**–**S10 Figs**). iPSC-derived motor neurons were positive for AchE

[111–113] and practically negative for DAT or GAD65/67 (**Fig 3C** and **S8–S10** Figs). The relative level of expression of these neuronal markers are summarized in **Table 1**.

## Only neurons are able to induce efficient HIV silencing in microglial cells

In order to address the specificity of neurons in inducing HIV latency in infected microglial cells, HC69 cells were co-cultured in the absence or presence of two non-neuronal cell lines, 293T cells and primary human foreskin fibroblasts (HFF) (**S11 Fig**). After co-culture for 24 h, 293 T cells and HFF failed to significantly decrease HIV expression in HC69 cells, as measured by flow cytometry (profiles (**S11A Fig**). Quantitation of three similar experiments (**S11B Fig**) showed that a slight decrease in HIV expression induced by HFF cells was not significant. Similarly, there was no significant toxicity detected in these co-cultures (**S11C Fig**), as measured by the resazurin assay. DEXA, a known inducer of HIV latency in microglial cells, was used as a positive control [39].

Similarly, neither iCort nor LUHMES neuronal cells were able to significantly reduce HIV expression in THP-1/HIV (A3) cells, a monocytic cell line [37], or Jurkat/HIV (2D10), a T-cell line [114] (**S12 Fig**). The ERK kinase inhibitor U0126, which has been shown to block HIV expression [115], was used as positive control in these experiments.

## Human iPSC-derived neurons can induce HIV latency in infected human primary microglia

In order to verify that primary neuronal cells are also able to induce HIV latency in infected primary microglial cells, we co-cultured both HIV-infected human primary microglial cells and iPSC-derived microglial cells with either primary neurons or iPSC-derived GABAergic cortical neurons. The primary and iPSC-derived microglial cells were infected with VSVG-HIV-GFP viruses using the protocols we established for immortalized human microglial cells [36–39]. To confirm latency in the primary microglial cells, the cells were cultured in presence DEXA, to induce HIV silencing, and reactivated with TNF-α, to induce any silenced proviruses in the culture (**Fig 4**). Both primary microglial (MG) (**Fig 4A**) and iPSC-induced microglial (iMG) (**Fig 4B**) cells can be readily infected with VSVG-HIV-GFP viruses. Approximately 40% of the MG cells and 30% of the iMG cells expressed HIV (GFP) at 72 h post infection (hpi). After treatment with DEXA for 24 h, virtually none of the infected MG or iMG cells expressed HIV. Treatment of the infected cells by TNF-α for 16 h increased the proportion of GFP+ cells in the culture by 25 to 50%.

The availability of HIV-infected primary MG and iMG cells permitted us to perform co-culture experiments in an entirely primary cell context using human primary neurons (Primary HN) and iCort neurons (**Fig 5**). The cells were cultured at a ratio of 50 neurons:6 microglia. Because of the limited numbers of cells that were available this experiment was performed semi-quantitatively using fluorescent microscopy. Nonetheless it is evident that Primary neurons were able to reduce HIV expression in both primary MG (**Fig 5A**) and iMG cells (**Fig 5B**) by more than 90%. The iCort neurons were able to reduce HIV expression both primary MG (**Fig 5A**) and iMG cells (**Fig 5B**) by more than 50%.

We have reported previously that HIV expression increases spontaneously in cultured hμglia/HIV cells due to autocrine production of TNF-α, potentially a consequence of the lack of neuronal inhibitory signals [39]. Because neurons were able to inhibit HIV expression in HC69 cells, at least in the short-term (**Figs 2 and 3**), we investigated whether neurons would also prevent spontaneous viral reactivation. A population of GFP⁻ cells were prepared by cell sorting from the mixed HC69 cell population, and co-cultured in the absence or presence of

**Table 1. Level of expression of neuronal markers in different neuronal cell types.**

| Marker | LUHMES | Cortical | Dopaminergic | Motor |
|---|---|---|---|---|
| beta-TUJ | +++ | ++ | +++ | +++ |
| MAP2 | +++ | +++ | +++ | +++ |
| CXCR3 | + | +++ | +++ | ++ |
| CD11b/c | - | - | - | - |
| GAD65/67 | +++ | +++ | + | + |
| DAT | +++ | + | +++ | + |
| AchE | +++ | + | ++ | +++ |

LUHMES-derived neurons. Viral transcription was then evaluated using flow cytometry and fluorescence microscopy (Fig 6).

In the absence of neurons, there was a 13% increase in GFP⁺ cells (from ~9% to ~22%) after 72 h, indicating the emergence of HIV from latency. In the presence of neurons, viral rebound was inhibited and GFP levels remained at ~8% as seen by the representative flow cytometry

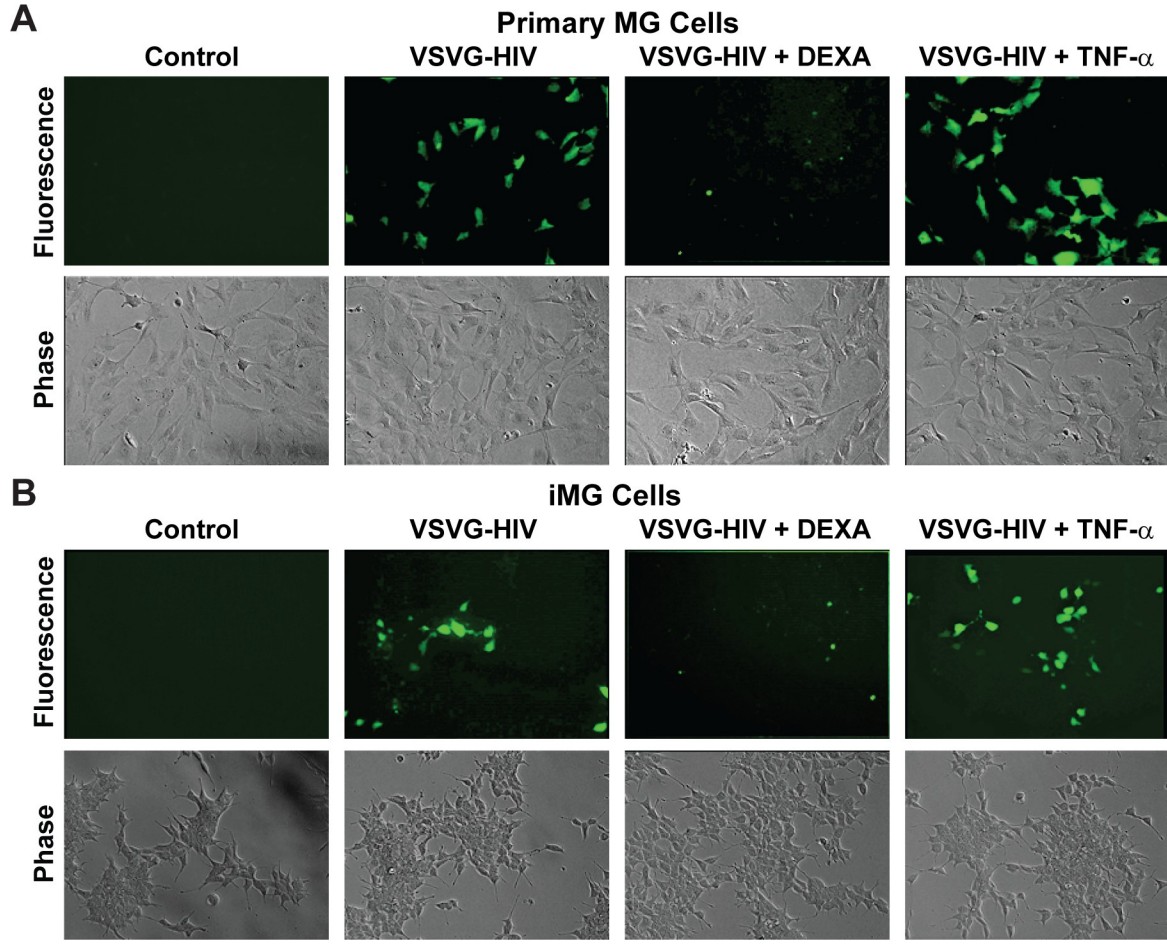

**Fig 4. Human primary and iPSC-derived microglia infected with VSVG-HIV-GFP can establish latent infections.** (**A**) Human primary microglial (MG) cells. (**B**) iPSC-derived microglial (iMG cells). Each cell type was infected with VSVG-HIV-GFP viruses for 72 h and GFP-expressing cells visualized by fluorescence microscopy. The infected cells were then either treated with DEXA (1 μM) for 24 h or with TNF-α (200 pg/mL) for 16 h and the proportion of GFP-expressing cells measured by fluorescence microscopy.

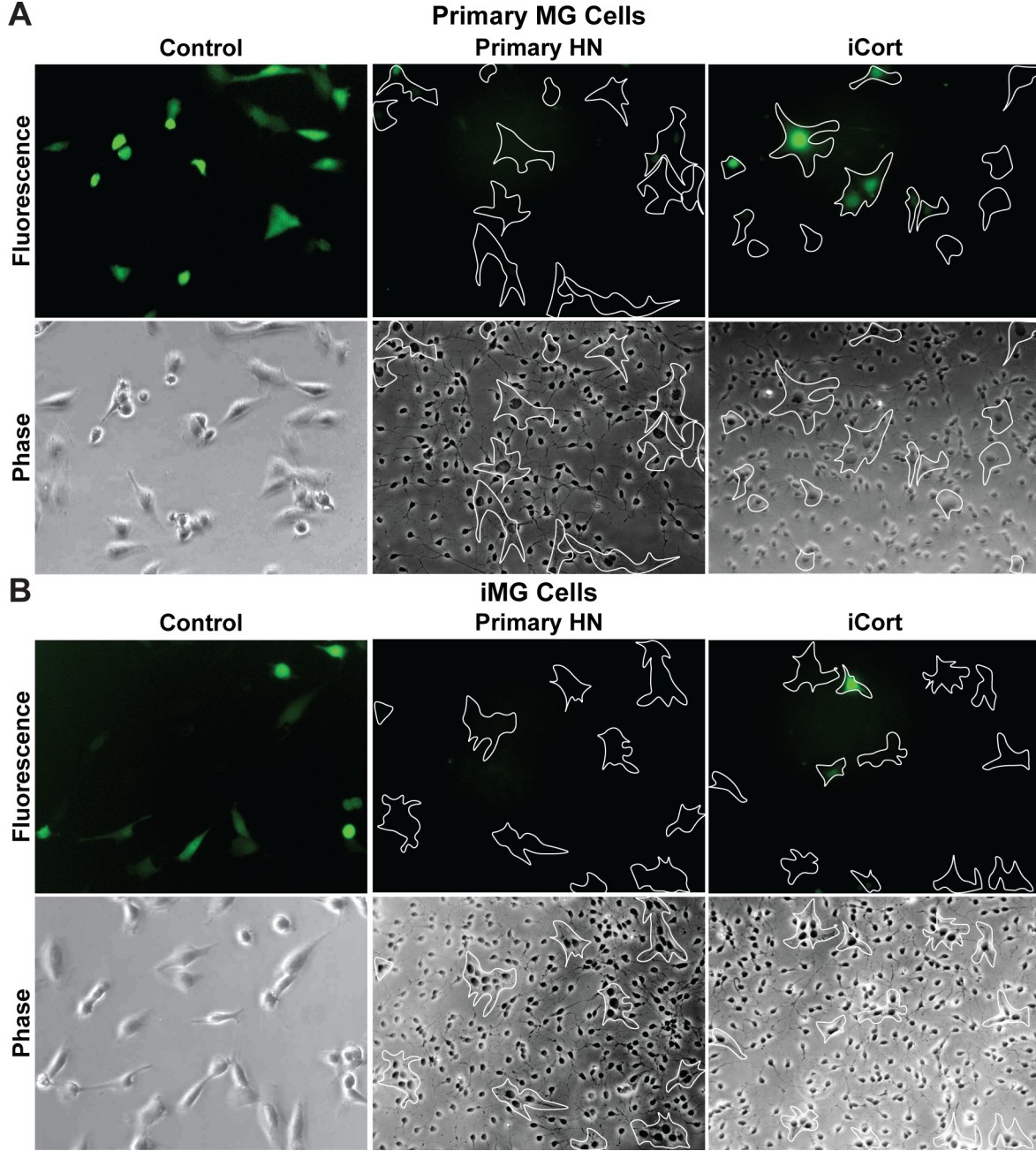

**Fig 5. Primary neurons silence HIV expression in primary microglia.** (**A**) Human primary microglial (MG) cells. (**B**) iPSC-derived microglial (iMG cells). For each cell type, $60 \times 10^3$ cells were plated in the absence or presence of $0.5 \times 10^6$ human primary neurons (Primary HN) or iPSC-derived GABAergic cortical neurons (iCort). HIV expression was evaluated after 24 h by fluorescence microscopy. Microglia identified by phase contrast microscopy are outlined by the white contours. Healthy neurons prevent spontaneous HIV reactivation in GFP⁻ cells.

profiles (**Fig 6A**) and microscopy (**Fig 6B**). Quantitation of multiple experiments (**Fig 6C**) showed that in the absence of neurons, HIV expression increased, on average, from $7.73 \pm 2.59\%$ at 24 h to $26.40 \pm 6.79\%$ ($p < 10^{-3}$) at 72 h, whereas in the presence of neurons, HIV expression remained at basal levels and did not vary significantly (N.S.). Neuronal viability, as measured using the resazurin reduction method, was unaffected after 24 h of co-culture

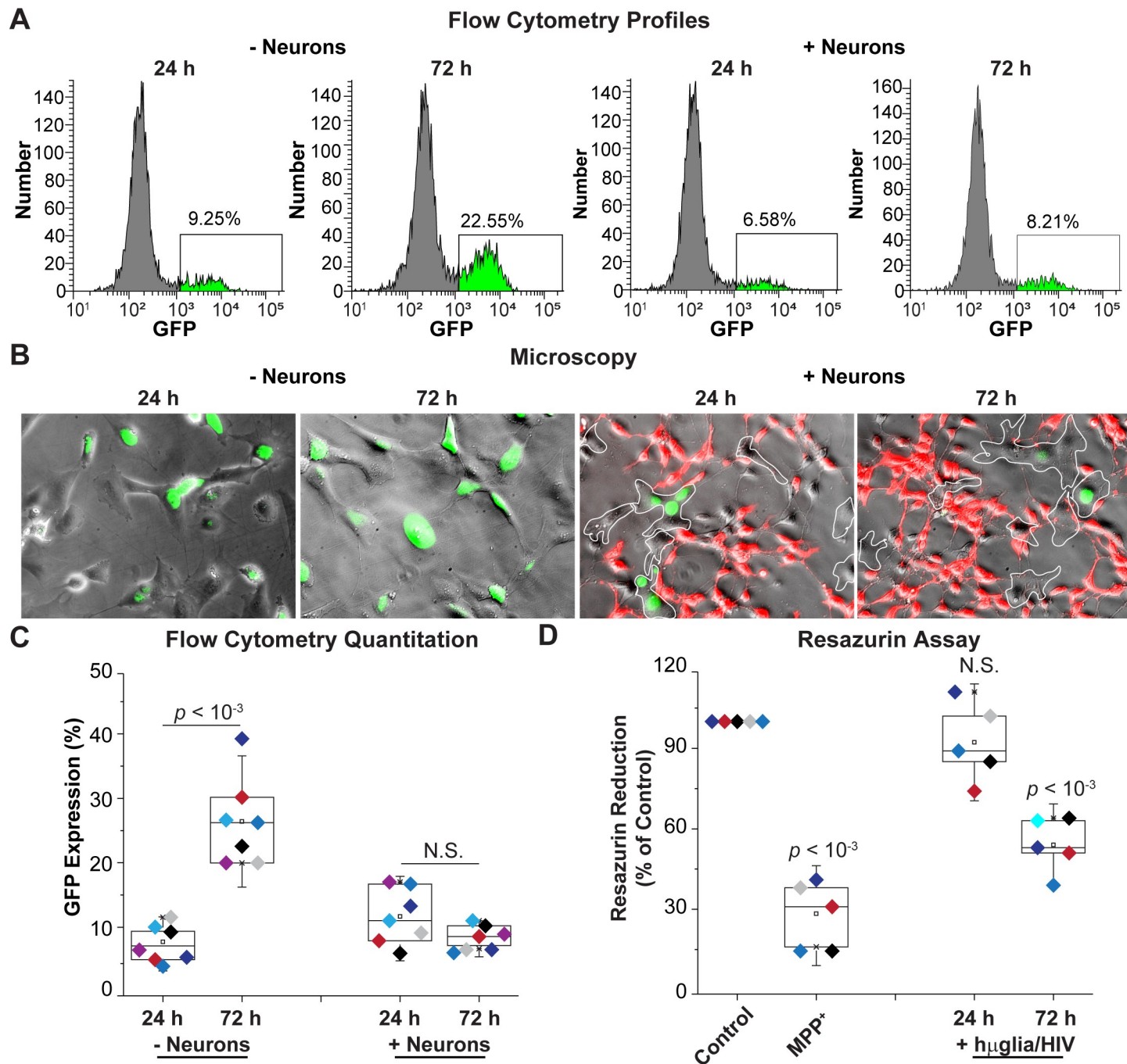

**Fig 6. Neurons prevent HIV emergence from latency.** (**A**) Flow cytometry profiles of representative single cultures. hμglia/HIV HC69 cells were sorted into a GFP⁻ cell population and cultured in the presence or absence of neurons. GFP expression was measured after 24 h or 72 h. (**B**) Microscopy of HC69 cells unexposed or exposed to neurons for 24 or 72 h. Microglial cells are outlined by a white dashed-line. (**C**) Quantitation of GFP expression. (**D**) Resazurin assay to evaluate neuronal viability. The resazurin reduction values (Y-axis) plotted are referenced to the control culture (neurons only), set at 100%. MPP⁺ was used a positive control for resazurin reduction. For both (C) and (D), the p-values of pair-sample t-tests of multiple experiments (n = number of independent samples) comparing the unexposed vs. the exposed cells are shown. N.S.: non-significant. Individual experimental series are color-coded.

with HC69 cells. There was a significant decrease in neuronal viability to $54.00 \pm 10.20\%$ ($p < 10^{-3}$) at 72 h (**Fig 6D**). For comparison, the neuronal toxin MPP⁺ decreased neuronal viability to $28.40 \pm 11.89\%$ ($p < 10^{-3}$). The increased neuronal toxicity at 72 h (**Fig 6D**) maybe

due to the partial inhibition of HIV expression, which may be sufficient to induce neuronal damage.

In order to rule-out the possibility that the observed block to HIV spontaneous reactivation in HC69 cells exposed to neurons was due to microglia-specific cell death or toxicity, we measured both the growth rate and the level of toxicity of the microglia population at 24 and 72 h. At both time-points, no statistically significant (N.S.) difference was observed in cell growth or toxicity level (Y-axis) between microglia cultured in the absence of neurons and microglia cultured in the presence of neurons (X-axis) (S13 Fig).

## Neuronal damage induces HIV expression

In the previous experiments, we showed that both LHUMES-derived and primary neurons can block HIV expression in microglial cells. To evaluate whether neuronal damage could reverse this repressive signal, HC69 cells sorted into GFP$^-$ and GFP$^+$ subpopulations and co-cultured for 24 h with either healthy neurons or neurons damaged by treatment with 0.05% trypsin [36] (Fig 7). Flow cytometry analysis of GFP expression in CD14$^+$ cells from a representative experiment showed that, as expected, co-culture with healthy neurons slightly decreased HIV expression in the GFP$^-$ cells from ~15% to ~10% (Fig 6A). By contrast, co-culture with damaged neurons had the opposite effect and increased GFP expression to ~19%. The healthy neurons strongly decreased HIV expression in GFP$^+$ cells (from ~78% to ~34%), whereas damaged neurons slightly increased it (from 78% to ~90%). Quantitation of multiple independent experiments (S14 Fig) indicated that healthy neurons reduced HIV expression in GFP$^-$ cells from 13.99 ± 3.14% to 11.05 ± 2.50% ($p < 10^{-3}$), whereas damaged neurons increased it up to 23.28 ± 1.68%; $p < 0.05$). In contrast, healthy neurons decreased HIV expression in GFP$^+$ cells (from 77.03 ± 9.84% to 44.99 ± 8.98% ($p = 0.003$)), whereas damaged neurons had no effect (88.11 ± 7.52%; N.S.).

To further investigate the effect of damaged neurons in modulating HIV expression in infected microglia, we systematically varied the ratio of healthy and damaged neurons. As shown in Fig 2 and Fig 7B, increasing the proportion of healthy neurons resulted in the progressive silencing of HIV and decreased from 44.14 ± 1.82% to a minimum of 11.19 ± 0.87%. By contrast when 0.6 x 10$^5$ HC69 cells were co-cultured in the presence of 0.5 x 10$^6$ damaged LUHMES-derived neurons, HIV expression increased from 44.14 ± 1.82% to 65.94 ± 4.06% ($p = 0.0011$). The inductive effect of the damaged neurons was ameliorated by the addition of healthy neurons. For example, when the co-cultures included 0.5 x 10$^3$ damaged and 0.3 x 10$^6$ healthy neurons, HIV expression decreased to 46.19 ± 0.41%. Similarly, when the co-cultures included 0.5 x 10$^3$ damaged and 0.5 x 10$^6$ healthy neurons, HIV expression decreased to 29.15 ± 7.88%.

In a complementary set of experiments (Fig 7C), 0.6 x 10$^5$ HC69 cells were co-cultured for 24 h with different ratios of damaged (D) and healthy (H) neurons at a fixed concentration of 0.5 x 10$^6$ neurons. In the presence of 0.5 x 10$^6$ damaged neurons, HIV expression increased from 34.31 ± 4.55% to 59.65 ± 9.37% ($p = 0.026$), consistent with the previous results. As the proportion of healthy neurons in the co-culture increased and the number of damaged neurons decreased, HIV was progressively silenced. For example, at a ratio of 3:2 damaged to healthy neurons, HIV expression remained unchanged (32.76 ± 3.82%) compared to the control microglial cells (Day 0). At a ratio of 2:3 damaged to healthy neurons, HIV expression significantly decreased 24.97 ± 4.66% ($p = 0.040$). At a 1:4 ratio, HIV expression decreased to 13.85 ± 2.31% ($p = 0.027$), which was close in value to 10.99 ± 2.06% ($p = 0.025$) seen in the absence of damaged neurons. Thus, damaged neurons present microglial activation signals that enhanced HIV expression.

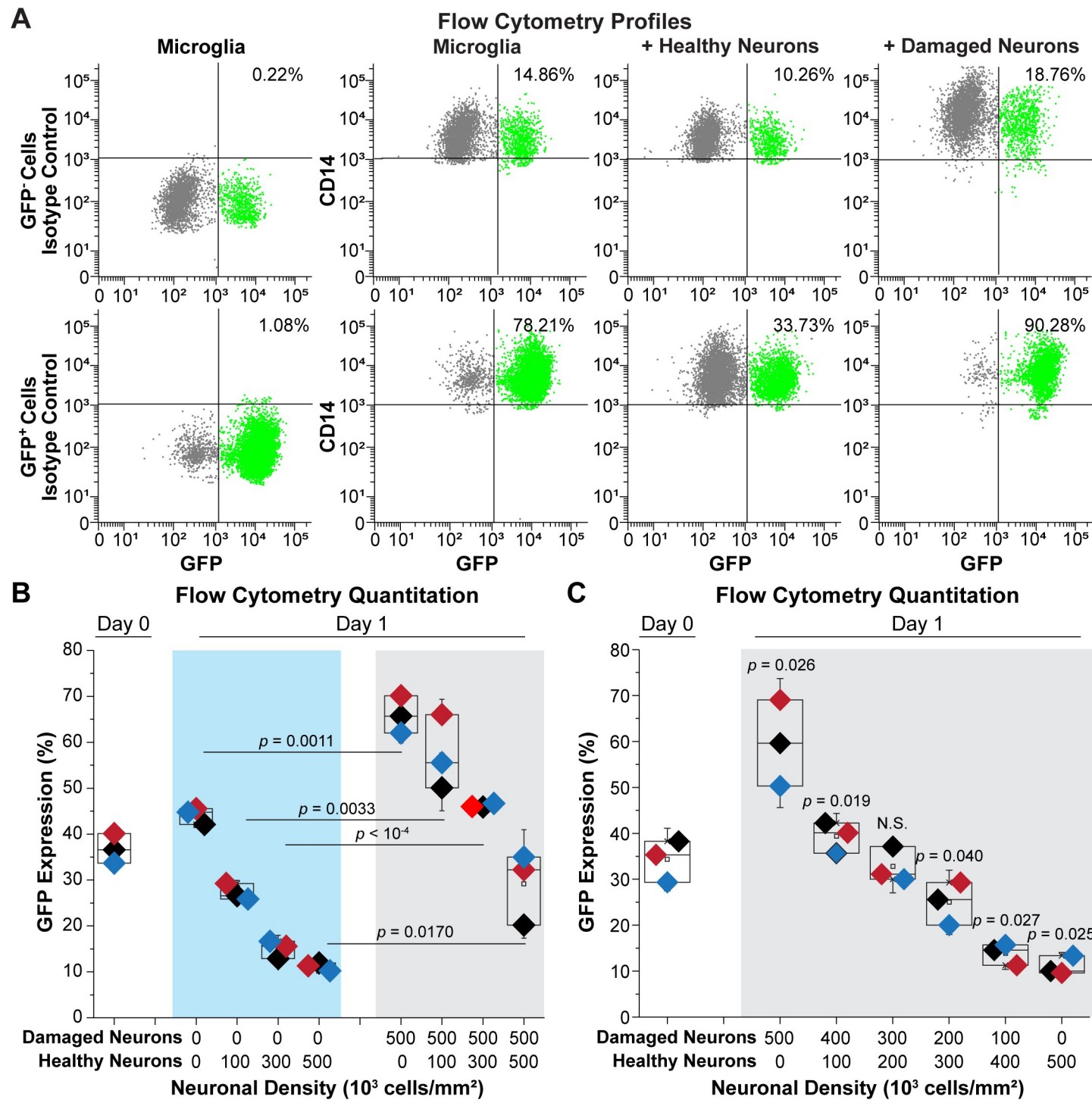

**Fig 7. Effect of healthy neurons vs. damaged neurons on HIV expression.** (**A**) Representative flow cytometry profiles. hμglia/HIV HC69 cells were sorted into GFP-
and GFP+ cells. Each population was expanded for 48 h prior to collection and co-cultured with either healthy neurons or damaged neurons at a ratio of 50:6. GFP (X-
axis) and CD14 (Y-axis) expression, and the *percentage* of GFP+ cells that were CD14+ is shown. Isotype controls for the anti-CD14 antibody were performed for both the
GFP+ and GFP- populations (left). (**B**) Quantitation of GFP expression. $60 \times 10^3$ HC69 cells were co-cultured in the presence of an increasing number of healthy (H)
neurons in the absence damaged (H) neurons (X-axis) for 24 h prior to measuring GFP expression (Y-axis) (left). In parallel experiments, microglial cells were co-
cultured with increasing numbers of healthy (H) neurons in the presence of $500 \times 10^3$ damaged (D) neurons (right). (**C**) Quantitation of GFP expression. $60 \times 10^3$ HC69
cells were co-cultured in the presence of $500 \times 10^3$ total neurons at the indicated ratios of damaged (D) to healthy (H) neurons (X-axis) for 24 h prior to measuring GFP
expression (Y-axis). Diamonds of similar color represent an individual experimental series. (*n* = number of individual samples). The *p*-values of paired-sample *t*-tests
comparing the unexposed vs. exposed cells are shown. N.S.; non-*s*ignificant.

## HIV expression rebounds in longer-term cultures with neurons

We next examined the effect of neurons on HIV expression in HC69 cells over a culture period of 4 days. A co-culture of LUHMES-derived neurons and GFP⁺ HC69 cells was established, and HIV expression and neuronal viability were monitored after 24, 72, and 96 h by microscopy and flow cytometry. As described above (**Figs 2 and 3**), neurons induced HIV to enter latency in a significant population of HC69 HIV-expressing cells after 24 h, as demonstrated by reductions in the number of green, GFP⁺ cells (**Fig 8A & 8B**). At 24 h, neuronal viability, as measured by resazurin reduction, was comparable to time zero (**Fig 8B**; ~100%). After 72 h, the number of GFP⁺ cells increased from ~30% to ~50% (**Fig 8B**), and this correlated with a reduction of neuronal viability (resazurin ~65%) and the relative proportion of live neurons counted (down to ~50%). By 96 h, the proportion of GFP⁺ cells increased reaching ~60% (**Fig 8B**), while resazurin reduced the proportion of neurons to ~45% and percent of live neurons to ~30% (**Fig 8B**).

## HIV expression in microglial cells induces neuronal damage

In order to demonstrate that the neuronal damage in the longer-term co-culture experiments was, at least in part, the result of the HIV expression, LUHMES-derived neurons were co-cultured (10:1 ratio) in the absence or presence of either C20 cells (the uninfected parental cell line) or HC20 cells (a mixed population of C20 cells infected with the HIV reporter) [36, 37]. After 96 h, neurons in the co-cultures were detected by immunocytochemistry with either anti-beta-TUJ antibody or anti-MAP2 antibody (Red), cultures were then stained with DAPI (Blue) (**Fig 9**). The green (GFP) cells are HC20 cells induced to express HIV. In the cultures with HC20 cells, there was relatively little reduction in the number of beta-TUJ positive cells compared to control C20 cells, indicating preservation of axons. By contrast, there was a marked reduction in intact dendrites as assessed by reductions in microtubule associated protein 2 (MAP2) immunoreactivity in neurons exposed to HIV-infected microglial cells, indicating extensive dendritic pruning.

Western blot analyses were also conducted to assess the extent of neuronal damage (**S15 Fig**). In these experiments, LUHMES-derived neurons were co-cultured in the absence or presence of C20 cells or HC69 cells, a clonal derivative of HC20 cells [36]. To maximize HIV expression, the HC69 cells were pre-treated with TNF-α or poly (I:C), a TLR3 agonist that mimics the effects of circulating bacterial rRNA and potently induces HIV by activation of IRF3 [37]. High levels of expression of α-amino-3-hydroxy-5-methyl-4-isoxazolepropionic acid receptor (AMPA receptor) [116–118], increased the phosphorylation of MAP2 (p-MAP2) [119, 120] and decreased phosphorylated synapsin I (p-synapsin) [121], which are both associated with neurodegeneration. Analysis of representative western blots showed no statistically significant (N.S.) difference between the effect of pre-activated C20 and pre-activated HC69 cells (**S15B Fig**), despite apparent increases in the expression of AMPA receptors in the presence of pre-activated microglia (**S15A Fig**). p(S136)-MAP2 levels increased more strongly with both TNF-α- and poly (I:C)-pre-activated HC69 than with similarly activated C20 cells (**S15A Fig**). p(S136)-MAP2 levels also showed significantly greater induction with TNF-α-pre-activated HC69 cells than with TNF-α-pre-activated C20 cells (*p* = 0.027). Similarly, poly (I:C)-pre-activated HC69 showed higher p(S136)-MAP2 levels than poly (I:C)-pre-activated C20 (*p* = 0.020). In contrast, p-synapsin levels (**S15A Fig**), were significantly lower in neurons exposed to TNF-α-pre-activated HC69 compared to C20 cells (*p* < 0.0001), but not altered when comparing neurons exposed to poly (I:C)-pre-activated HC69 or C20 cells (N.S.). Interestingly, based on the resazurin reduction method (**S15C Fig**), TNF-α- or poly (I:C)-pre-

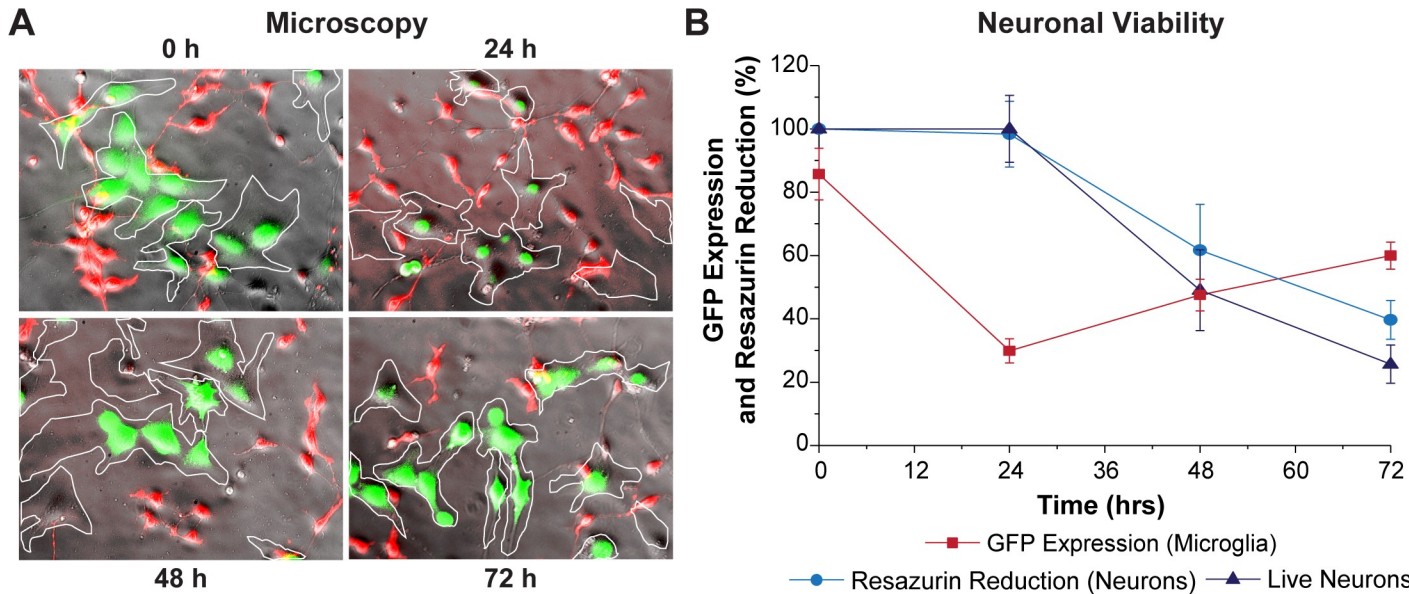

**Fig 8. HIV expression rebounds in microglia exposed to neurons.** (**A**) Super-imposed phase contrast and fluorescence images of HC69 GFP⁺ cells exposed to neurons for 24, 72, or 96 h. Microglial cells are outlined by the white contours. (**B**) Quantitation of GFP expression (red), relative resazurin reduction (blue), and relative live neuron counts (purple), Y-axis, vs. Time, X-axis. Error bars: standard deviation (*n* = 3).

activated HC69 cells induced significantly more neuronal damage than TNF-α-pre-activated C20 cells (*p* = 0.048 and *p* = 0.030, respectively).

We also observed extensive neuronal damage following HIV induction when LUHMES-derived neurons were co-cultured (10:1 ratio) with rat-derived microglia CHME-5 cells [36]. Co-culture experiments were performed using either uninfected CHME-5 cells, cells that were latently-infected with the HIV reporter, or latently infected cells that were induced by treatment with TNF-α (100 ng/mL) (**S16A Fig**). After 48 h, the co-cultures were immunostained with anti-MAP2 antibody (Red) and the nuclei were visualized with DAPI (Blue). Green (GFP) cells are activated CHME-5/HIV. When neurons were co-cultured with uninfected CHME-5 cells, activation with TNF-α induced relatively little neuronal injury, as evidence by the reduced number of MAP2 immunoreactive neuronal dendrites. However, when neurons were co-cultured with CHME-5/HIV cells, addition of TNF-α greatly exacerbated neuronal damage, as evidenced by marked dendritic pruning compared to CHME-5 cells in the presence of TNF-α.

A similar effect was observed when we measured the effect of TNF-α-pre-activated CHME-5 and CHME-5/HIV cells on neuronal viability by measuring the relative number of healthy neurons visually and manually counted in time-lapse experiments. CHME-5/HIV cells, which are a rat cell model for HIV latency [36], decreased neuronal viability from 100% to ~35% (pink), compared to the co-cultured with CHME-5 cells (down to ~75%; blue) or neurons alone (black), where viability remained above 90% for the duration of the experiment (**S16B Fig**).

## METH induces HIV expression in hμglia/HIV cells in a σ1R-dependent manner

We have shown previously that high concentrations of METH induce HIV reactivation in rat CHME-5/HIV cells [13]. To test the ability of METH to reactivate HIV in hμglia/HIV cells,

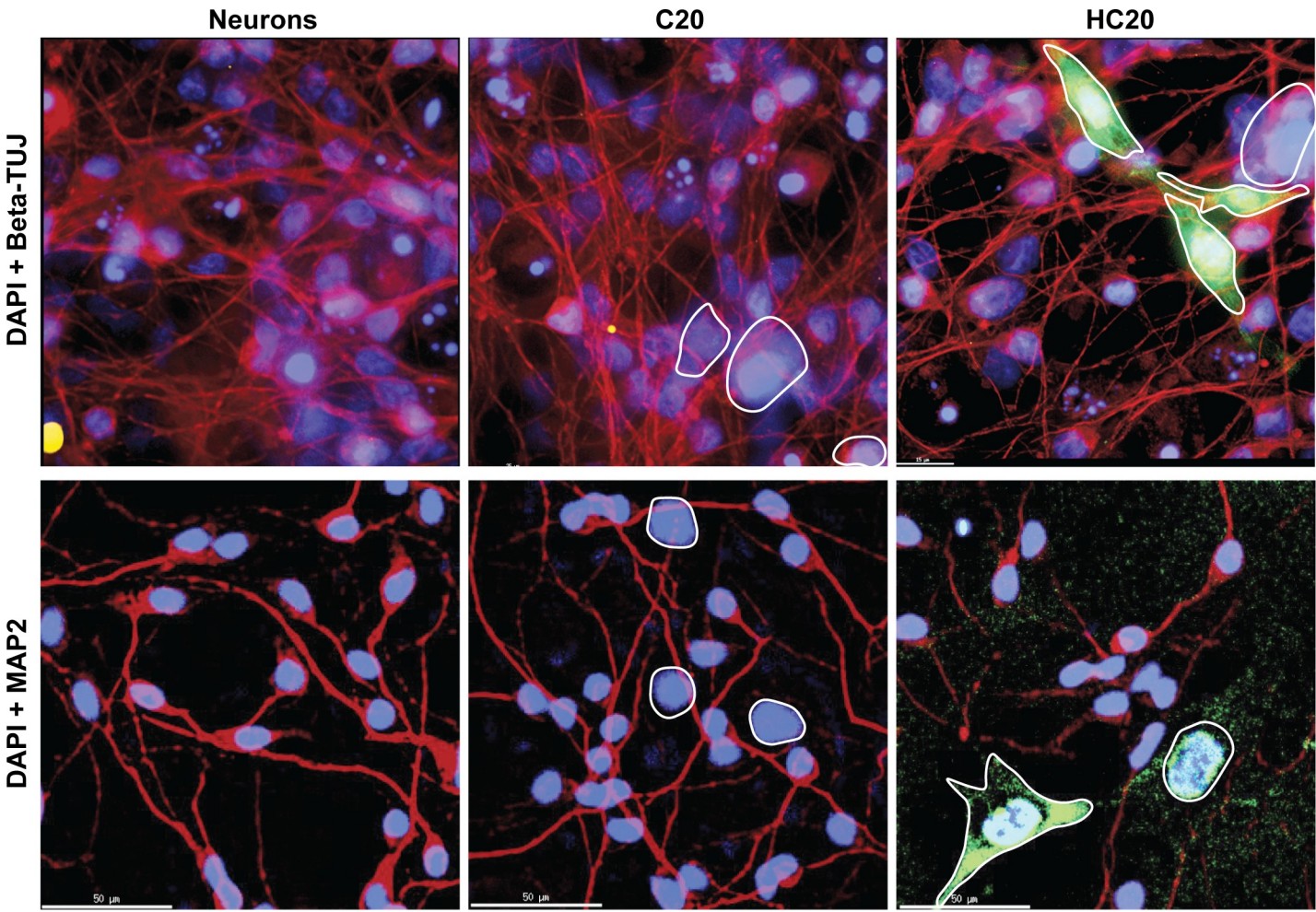

**Fig 9. Activated human microglia/HIV induce neuronal damage.** LUHMES-derived neurons were co-cultured with microglia clone C20 or mixed population of cells that had been infected with HIV (HC20). Top: The neurons or co-cultures were stained with anti-beta-TUJ antibody (Red). Bottom: Neurons were co-cultured with C20 or HC20 cells and stained with anti-MAP2 (Red). Green: GFP expression in activated HC20 cells. Red: Alexa Fluor 488 antibodies were used as secondary antibodies. Blue: DAPI stained cell nuclei.

HC69 cells were incubated with 50 to 500 μM METH (**Fig 10**). These concentrations are in excess of the 100 nM to 10 μM METH levels seen in chronic drug abusers [122], but were chosen to evaluate the maximal level of METH that did not induce cytotoxicity. At the high METH concentrations, over a 96 h time course there was a progressive increase in HIV expression, as demonstrated by the levels of GFP+ cells in the population (**Fig 10A**).

HIV was not reactivated during the first two days at any of the tested drug concentrations but became evident after 3 days at METH concentrations above 100 μM (**Fig 10A**). At 50 μM (blue circles and line), there was no effect on HIV reactivation (**Fig 10A**) or cell viability as assessed by propidium iodide (PI) exclusion (**Fig 10B**) compared to untreated controls during the course of the 5 days (0 μM; red squares and line). At 100 μM (yellow triangles and line), HIV reactivation was higher than in controls from day 3 (**Fig 10A**), but viability was somewhat compromised compared to untreated cells (**Fig 10B**; ~ 91% vs. ~ 86% at day 3, ~ 89% vs. 81% at day 4, and ~ 93% vs. 73% at day 5). At 300 μM and 500 μM METH, HIV reactivation was maximal (**Fig 10A**) but cell viability was compromised. Flow cytometry histograms for cells treated with 300 μM METH from a similar experiment are shown in **S17 Fig**. No activation

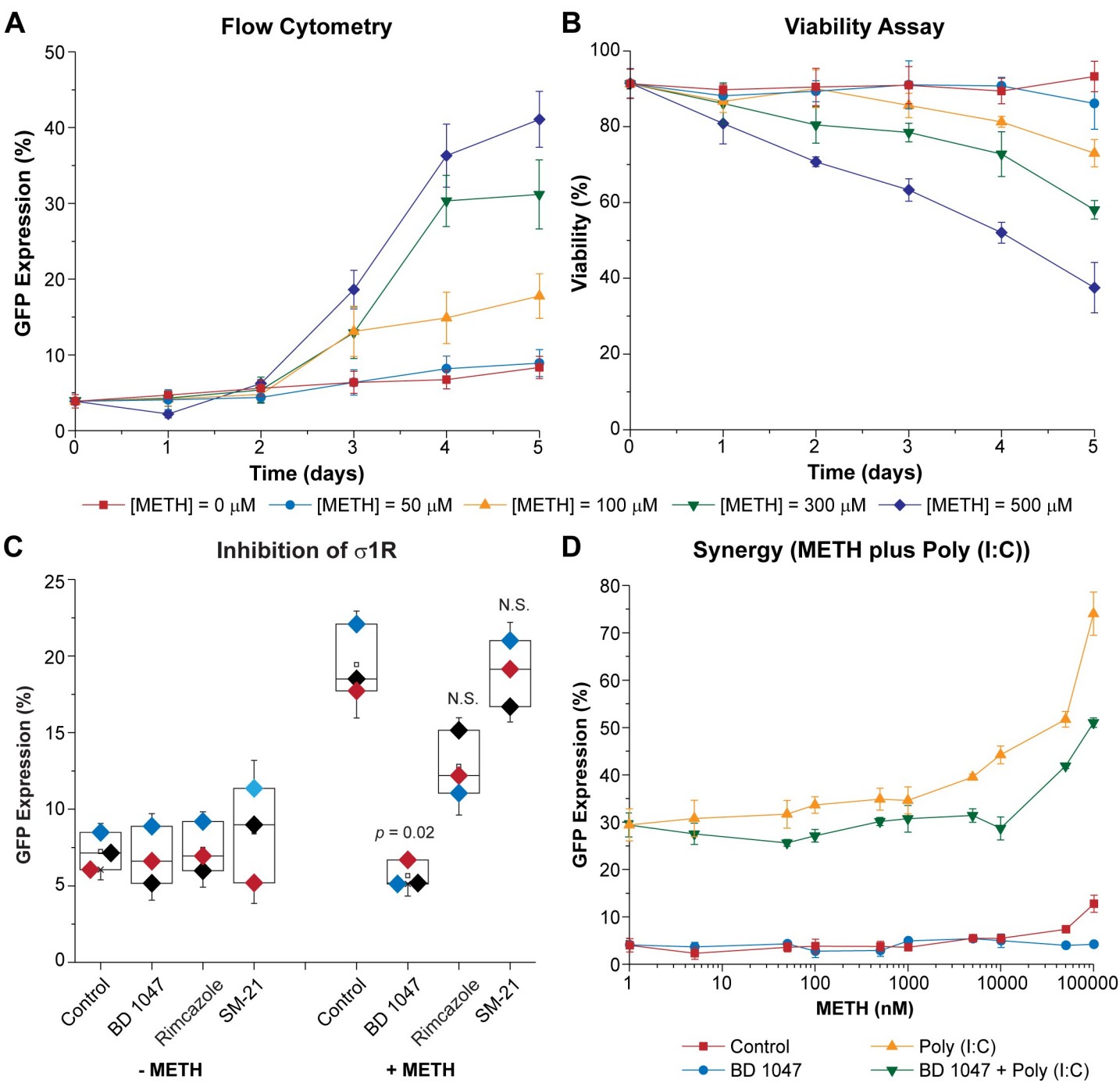

**Fig 10. METH-mediated reactivation of HIV.** (**A**) Effect of METH on HIV reactivation. (**B**) Cell viability. HC69 cells were incubated for 1, 2, 3, 4 or 5 days (X-axis) with indicated concentrations of METH prior to flow cytometry analysis and PI exclusion cell viability assay (Y-axis). Error bars represent standard deviations of three or more experiments. (**C**) σ1R mediates METH effect on HIV reactivation. HC69 cells were either untreated (Control) or treated with BD1047, rimcazole, or SM-21 prior to exposure to 300 μM of METH (+ METH) or untreated (- METH). Diamonds of similar color represent an individual experimental series. (*n* = number of individual samples). The *p*-values of pair-sample t-tests comparing the unexposed vs. the exposed cells are shown. N.S.: non-*s*ignificant. (**D**) METH sensitizes hμglia for poly (I:C)-mediated HIV reactivation at low doses. HC69 cells were treated with increasing concentrations of METH (Control; 0, 1, 5, 50, 100 and 500 nM, and 1, 5, 10, 50 and 100 μM; X-axis, log scale) for 72 h prior to exposure to poly (I:C) (50 ng/mL). Parallel experiments were performed in the presence of BD1047 (10 μM). Error bars represent standard deviation of three or more experiments.

was observed after 24 h (from ~7% to ~4% GFP$^+$) (**S17B Fig**), compared to the untreated controls; however, by 96 h, ~32% of the cells became GFP$^+$ (**S17B Fig**) compared to ~ 9% in the control cells (**S17A Fig**). In control experiments, potent proviral reactivation was seen at each time point using TNF-α (at 100 pg/mL; ~ 90%) (**S17C Fig**) or poly(I:C) (at 100 ng/mL; ~68%) (**Fig 8D**).

Several studies have suggested that METH exhibits significant affinity for the σ1R. For example, BD1047, a specific inhibitor of the endoplasmic membrane-bound σ1R, was found to reduce neuronal injury in METH-exposed hippocampus [123]. More recently, it has been reported that σ1R antagonists attenuated METH-induced hyperactivity and neurotoxicity [124], and that σ1R is involved in METH-mediated microglial activation [125]. HC69 cells were exposed to METH at 300 μM, the dose found to be more effective at inducing HIV reactivation with low cell toxicity (**Fig 10A & 10B**), in either the absence or presence of BD1047 (10 μM), a σ1R antagonist [126], rimcazole (10 μM), a σ1R and σ2R antagonist [127], and SM-21 maleate (10 μM), a σ2R antagonist [128]. As shown in **Fig 10C**, BD1047, which by itself had no effect on HIV expression compared to controls, inhibited METH-mediated HIV reactivation from 19.44 ± 1.22% down to 5.66 ± 0.89% ($p = 0.02$). Rimcazole did not significantly reduce HIV expression and SM-21 maleate had no apparent effect.

We reasoned that in HIV-infected individuals who utilize METH, they are also invariably exposed to chronically increased levels of microbial products, such as LPS and bacterial rRNA, and proinflammatory cytokines due to damage to the gut permeability barrier [129, 130]. To mimic these conditions, we evaluated the impact of METH on HIV expression in microglial cells in the presence of two pro-inflammatory agents, TNF-α and poly (I:C). As shown in **S18 Fig**, pre-treatment by 100 μM METH for 72 h sensitized HC69 cells to HIV reactivation in response to sub-saturating concentrations of poly(I:C) (50 ng/mL), but not for sub-saturating concentrations TNF-α (20 pg/mL). In this experiment, METH combined with poly(I:C) reactivated ~77% of the cells, compared to ~38% of the cells after treatment by poly(I:C) alone ($p = 0.01$).

Because of the synergy observed between METH and poly (I:C), we performed another similar experiments using a broad range of METH concentrations spanning the exposure doses seen in drug abusers (1 nM to 100 μM) in the absence or presence of 50 ng/mL of poly (I:C). As a control for the impact of METH on the cells, parallel samples were treated with BD1047. As shown in **Fig 10D**, even at nM concentrations, METH (red squares and line) sensitized HC69 cells for poly (I:C)-mediated HIV reactivation (yellow triangles and line; from ~28% to ~29% at 1 nM, ~31% at 5 nM, ~32% at 50 nM, 34% at 100 nM, and 35% at 500 nM). At higher METH doses, the sensitization effect was more pronounced (to ~35% at 1 μM, ~40% at 5 μM, ~44% at 10 μM, ~52% at 50 μM, and 74 at 100 μM). As expected, the inhibitor BD1047 (10 μM) abrogated the effect of METH (**Fig 10D**).

## METH induces HIV expression in neuronal mixed-glial co-cultures

We hypothesized that METH-induced neurodegeneration is exacerbated due to a combination of its direct activation effect on HIV expression and indirect effects due to neuronal damage. To test our hypothesis, HC69-iCort neuronal co-cultures, as described above (**Fig 3**), were exposed to METH 100 nM, a dose shown to slightly synergized with poly(I:C) for HIV reactivation in HC69 cells (**Fig 10D**), in the absence or presence of poly(I:C) 50 ng/mL for 96 h prior to evaluating GFP expression and toxicity. In the presence of METH, HIV reactivated in a significant proportion of HC69 cells (**Fig 11A**). GFP expression increased from 4.27 ± 1.88% to 10.22 ± 2.06%, $p < 0.001$ (**Fig 11B**), whereas toxicity decreased from 100% to 77.60 ± 5.01%, $p = 0.016$ (**Fig 11C**). In the absence of neurons (**Fig 10D**), we did not observe this effect suggesting that METH-mediated neuronal damage, evidenced by the increased dendritic

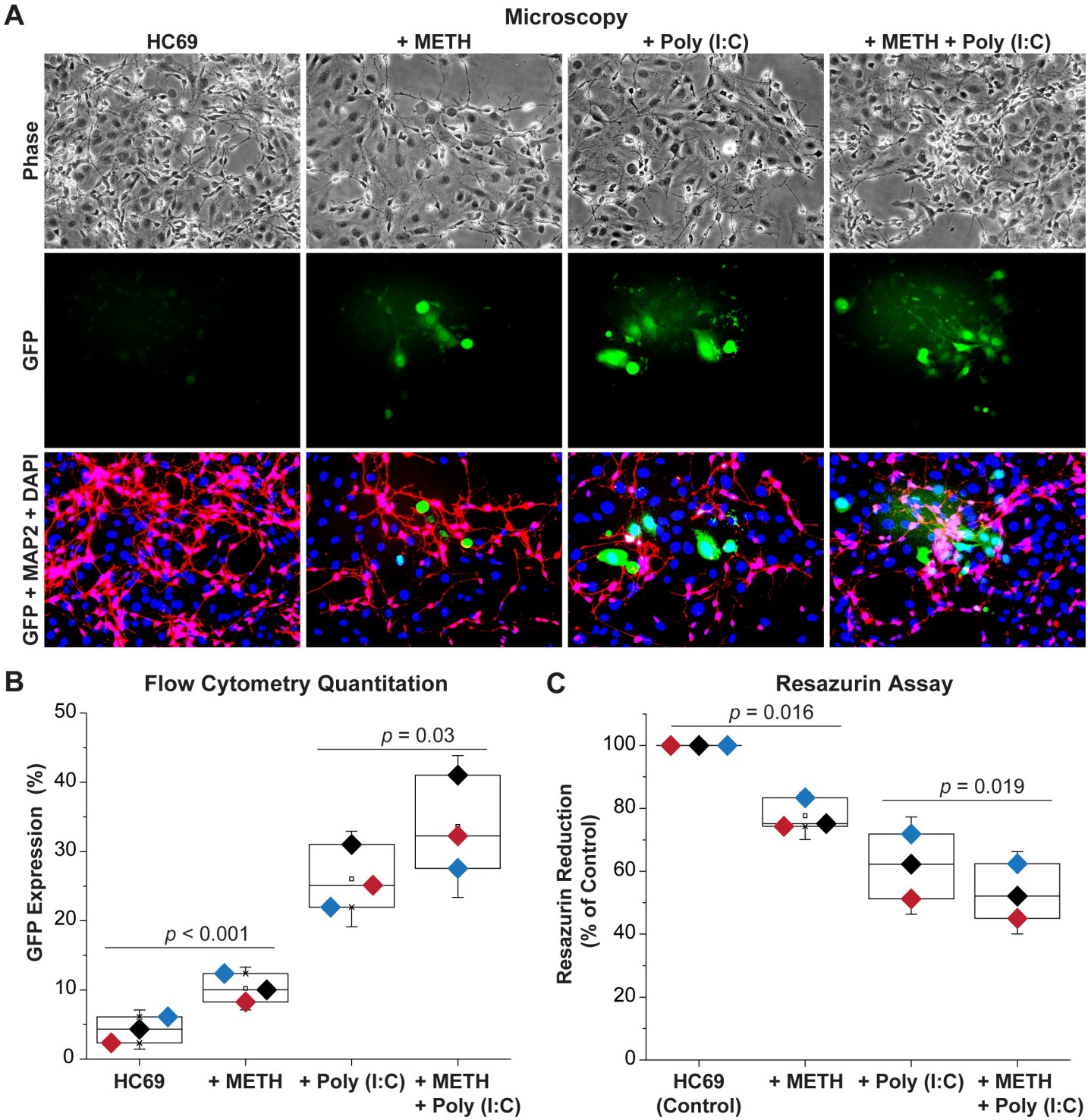

**Fig 11. Exposure of co-cultures to combinations of METH- and Poly (I:C) induce extensive neuronal damage.** (**A**) Microscopy of HC69 cells co-cultured with iCort neurons for 96 h in the presence and absence of 100 nM METH and 50 ng/ml Poly (I:C). Top: Phase contrast image. Middle: GFP positive microglial cells. Bottom: Neurons were stained with anti-MAP2 (Red). Green: GFP expression in activated HC69 cells. Red: Alexa Fluor 488 antibodies were used as secondary antibodies. Blue: DAPI stained cell nuclei. (**B**) Induction of GFP expression in HC69 cells co-cultured with iCort neurons for 96 h and then measured by flow cytometry. Cells were treated with 100 nM METH or 50 ng/ml Poly (I:C) or a combination of 100 nM METH or 50 ng/ml Poly (I:C). Error bars represent standard deviations of three experiments. (**C**) Resazurin assay to evaluate neuronal viability. The resazurin reduction values (Y-axis) plotted are referenced to the control culture (untreated HC69 cells), set at 100%. The $p$-values of pair-sample t-tests of multiple experiments comparing the control vs. the METH exposed cells and Poly (I:C) treated cells vs. the METH plus Poly (I:C) treated cells are shown. Error bars represent standard deviation of three experiments.

fragmentation and toxicity (Fig 11A & 11C), induces HIV emergence from latency. Poly (I:C), by itself, also increased the number of HIV-expressing HC69 cells in co-cultures with GFP expression increasing up to 26.04 ± 4.60% (Fig 11B), while increasing dendritic fragmentation (Fig 11A) and neurotoxicity with only 61.79 ± 10.30% surviving (Fig 11C). This result suggests that the neuronal toxicity induced by poly(I:C) is directly related to release of HIV and pro-inflammatory agents from the induced microglial cells. The combination of METH and poly(I:C) further significantly induced HIV expression and dendritic damage compared to cells treated with poly(I:C) alone (Fig 11A). The combination of METH and poly(I:C) increased GFP expression up to 33.61 ± 6.83% (Fig 11B, $p = 0.03$) and decreased resazurin reduction to 53.16 ± 8.73% (Fig 11C, $p = 0.019$).

We also established a co-culture between primary mouse neurons and microglial cells carrying the HIV reporter (HC69) or uninfected control (C20) cells. HC69 cells treated with 300 µM METH for 3 days displayed increased HIV expression when co-cultured with neurons compared to cultures not exposed to METH. As shown in S19 Fig, when neurons were co-cultured with HC69 cells, METH exposure induced reductions in dendritic diameter and there was evidence of increased dendritic fragmentation compared to C20 or HC69 cells in the absence of METH (S19 Fig).

Finally, in similar experiments using LUHMES-derived neurons co-cultured with either C20 or HC69 cells (S20 Fig), we used the resazurin method to assess neuronal viability after treatment of microglial cells with a variety of inflammatory or anti-inflammatory agents. Reactivation of HIV by TNF-α (20 pg/mL) or poly(I:C) (50 ng/mL) decreased neuronal viability by 15–25% in C20 cell co-cultures. In combination with METH, neuronal viability was further decreased (by ~ 30%) (S20A Fig). These neurotoxic effects were significantly enhanced when HC69 cells were used in the co-cultures. Neuronal viability was decreased down to ~ 60% after activation by TNF-α or poly(I:C) and, reduced to ~ 50% when METH was combined with TNF-α (similar to METH alone) or to ~ 30% with poly(I:C) (S20A Fig). In control experiments assessing neuronal damage in the absence of microglia, toxicity was slightly less than that observed in the co-cultures with C20 cells (S20A Fig). By contrast, the neurotoxin MPP$^+$ reduced neuronal viability to ~ 25%.

In comparison, dexamethasone (DEXA), which we have shown can potently repress HIV expression in microglial cells [39], was neuroprotective (S20B Fig). We attribute DEXA's neuroprotective effects to its ability to restrict HIV reactivation and thereby limits HIV-induced bystander neuronal injury [39].

In a parallel experiment, METH (300 µM) reduced neuronal viability by about 30% in C20 cell co-cultures ($p = 0.025$), compared to losses of about 50% in the HC69 cell co-cultures ($p = 0.005$) (S20A Fig). However, the selective σ1R antagonist BD1047, which by itself had no apparent effect on neuronal viability, largely negated the neurotoxic effects of METH in C20 and HC69 cell co-cultures (S20B Fig). This finding indicates that METH acts through specific σ1Rs to reactivate HIV and trigger neurotoxicity in neuronal-microglial co-cultures.

We conclude that METH can exacerbate the neuronal damage seen in the presence of HIV-infected microglial cells through a combination of direct neurotoxic effects that activate the HIV transcription in infected microglial cells and by potentiating HIV reactivation responses to pro-inflammatory agents.

## Discussion

### Microglial dysregulation by HIV results in synaptodendritic injury and neuronal losses

HAND remains a significant clinical problem as it manifests in approximately 30–50% of cART-treated patients, despite peripheral viral suppression [5]. Pathophysiologically, HAND

is driven by the sustained, low-level production of HIV proteins and proinflammatory toxins in microglia that result in synaptodendritic injury, reductions in CNS connectivity with accompanying neurobehavioral and cognitive declines [131–134]. The poor CNS penetration and consequently reduced effectiveness of certain ART drugs creates an environment where HIV replication is favored and has also been linked to the development of HAND [135].

In normal situations, activation of microglia in response to inflammatory stimuli is characterized by a transition from a resting state (M0 or M2 cells) to more activated cells (M1 cells). Although Ransohoff has argued that this model is an oversimplification [136], we believe it remains a useful broad classification of glial activation states. In contrast to other cells of the immune system, strict limitations on the activity of stimulated microglial cells are imposed to avoid an exaggerated response during infection and injuries. These mechanisms result both in the production of anti-inflammatory cytokines and inhibitory proteins, and attenuated production of pro-inflammatory cytokines through finely coordinated cell signaling and transcriptional programs such as the Nurr1/CoREST transrepression pathway [137, 138]. It is generally believed that over-activated microglia (constitutive M1 cells) exacerbate neuronal injury through the synthesis and secretion of cytotoxic factors, which increase excitotoxic synaptic transmission and damage healthy neurons [47, 48]. Therefore, microglia-mediated neurotoxicity, appears to be the result of excessive and uncontrolled stimulation [49, 50], and/or impaired functionality of intrinsic molecular mechanisms [51–53], which are likely to be further impaired as a consequence of HIV infection.

Neuronal-microglial communication is now realized to be far more dynamic than previously thought, and neurons play key roles in regulating microglial responses [139, 140]. For example, neurons regulate microglia through the CX3CL1 (expressed by neurons)/CX3CR1 (expressed in microglia) axis [141–145], and through the neuronal surface proteins CD200, CD47 and CD22, which bind to their cognate receptors CD200R, CD172 and CD45, respectively, on microglia [146].

Importantly, neuronal dysfunction in HAND does not correlate with the number of HIV-infected cells or viral antigens in CNS [82, 83], but rather with elevated inflammatory cytokine levels. These observations have led to some investigators to minimize the importance of viral persistence as part of the etiology of NeuroHIV [82, 84–86, 88].

## Microglial-neuronal crosstalk regulates HIV latency

*In-vitro* neuron-glia co-culture strategies have been widely used to gain an in-depth understanding of regulatory interactions between neurons and microglia [90–96]. Here, we describe a co-culture method between primarily LUHMES-derived neurons and immortalized human microglia infected with an HIV reporter construct. To confirm these observations in the most realistic cell systems available, we have also co-cultured primary or iPSC-derived neurons with primary or iPSC-derived microglia.

Using these systems, we have observed, unexpectedly, that neurons can silence HIV expression in infected microglia and prevent spontaneous reactivation of latent virus. Interestingly, iPSC-derived GABAergic cortical and dopaminergic neurons, but not motor neurons, were capable of mediating HIV silencing, recapitulating the effect of LUHMES-derived neurons.

One plausible explanation for the neuronal specificity regulating HIV silencing is that dopaminergic and GABAergic cortical neurons, but not cholinergic motor neurons, are able to synthesize and release anti-inflammatory compounds such as glucocorticoids (or neurosteroids). Glucocorticoids (GC) play a key role in countervailing inflammation in the CNS and can protect the brain against excessive innate immune responsiveness [39, 147–152]. The hypothalamic-pituitary-adrenal (HPA) axis rapidly increases the release of circulating glucocorticoids

in response to various cytokines such as TNF-α and IL-1β. Consequently, by binding to the glucocorticoid receptor (GR), GCs and their analogues such as dexamethasone (DEXA) repress transcription of the inflammatory genes by blocking recruitment of NF-κB and AP-1 to their promoters [39, 153]. Consistent with these mechanisms, activation of the GR by DEXA in hμglia/HIV cells promotes HIV latency because it reduces microglial pro-inflammatory responses, whereas down regulation of the GC receptor by shRNA leads to HIV reactivation [39]. Other possible explanations include the loss of essential neuronal proteins that promote microglial inactivation. Further studies are warranted to determine the level of expression of C200, CD47, CD22, and CX3CL1 on GABAergic cortical, dopaminergic and motor neurons, which could further explain the selectivity of GABAergic cortical and dopaminergic neurons to silencing HIV.

While healthy neurons can promote HIV silencing, damaged neurons reactivate microglial cells and enhance HIV transcription, resulting in further neuronal deterioration and the increased expression of damage-associated molecular patterns (DAMPs). Therefore, we hypothesize that the "off" signals that keep microglia in check [146] also inhibit viral expression in infected microglia. Thus, neuronal DAMPs can trigger low, persistent levels of viral expression/reactivation resulting in sustained cycles of microglial activation, inflammation, HIV reactivation, and the production of more DAMPS from injured bystander neurons.

In our co-culture experiments, we have observed that in after several days in culture with neurons HIV-expressing infected microglia induce excessive neuronal damage due to initiation of a vicious cycle of enhanced neuronal damage leading to enhanced microglial activation. After an initial period when the healthy neurons can repress HIV reactivation, a vicious cycle becomes established where damaged neurons induce HIV expression, leading to further neuronal damage. The results presented here, and our previous work, support the hypothesis that HIV expression in infected microglia leads to an enhanced response to inflammation and periodic cycles of neuronal damage [36, 37, 39] (**Fig 12**). One implication of this hypothesis for the etiology of HAND is that multiple pro-inflammatory episodes that are damaging to neurons lead to increased HIV expression and creation of a constitutively M1 state for microglial cells in the brain.

Primary human microglia are efficiently infected by replication competent R5 HIV (AD8gNef-GFP), and primary macaque microglia with replication competent SIV 17E-Fr particles [36, 154]. Unfortunately these cells loose the CD4 receptor upon passaging making it technically impossible to obtain sufficient susceptible primary microglia to perform studies of the type shown here. Therefore, for most of our studies we have relied on the HC69 clonal population of immortalized human microglial cells bearing an HIV construct (hμglia/HIV) [36, 37, 39] as a proxy. HC69 bears an HIV construct that lacks Gag, Pol and other accessory viral proteins. We have also previously shown that the proviral integration site of the HC69 cells was sequenced and located within the host genome, demonstrating that HC69 is a single integrant and, as expected from the extensive studies characterizing HIV proviral integration sites, the provirus was located in the introns of host genes [39]. There is extensive evidence that these reporters, which we used extensively to study HIV transcription in T-cell systems [155–157] and microglial cells [36–39] accurately reflect the transcriptional state of the virus since GFP expression is strictly dependent upon Tat activation in this system. We therefore believe that the neuronal silencing we have observed is the result of blocking of signaling pathways that activate HIV expression. It is also important to note that even microglial cells carrying defective proviruses can induce extensive neuronal damage once they become activated. The degree of neurotoxicity is likely to increase in the context of replication competent viruses.

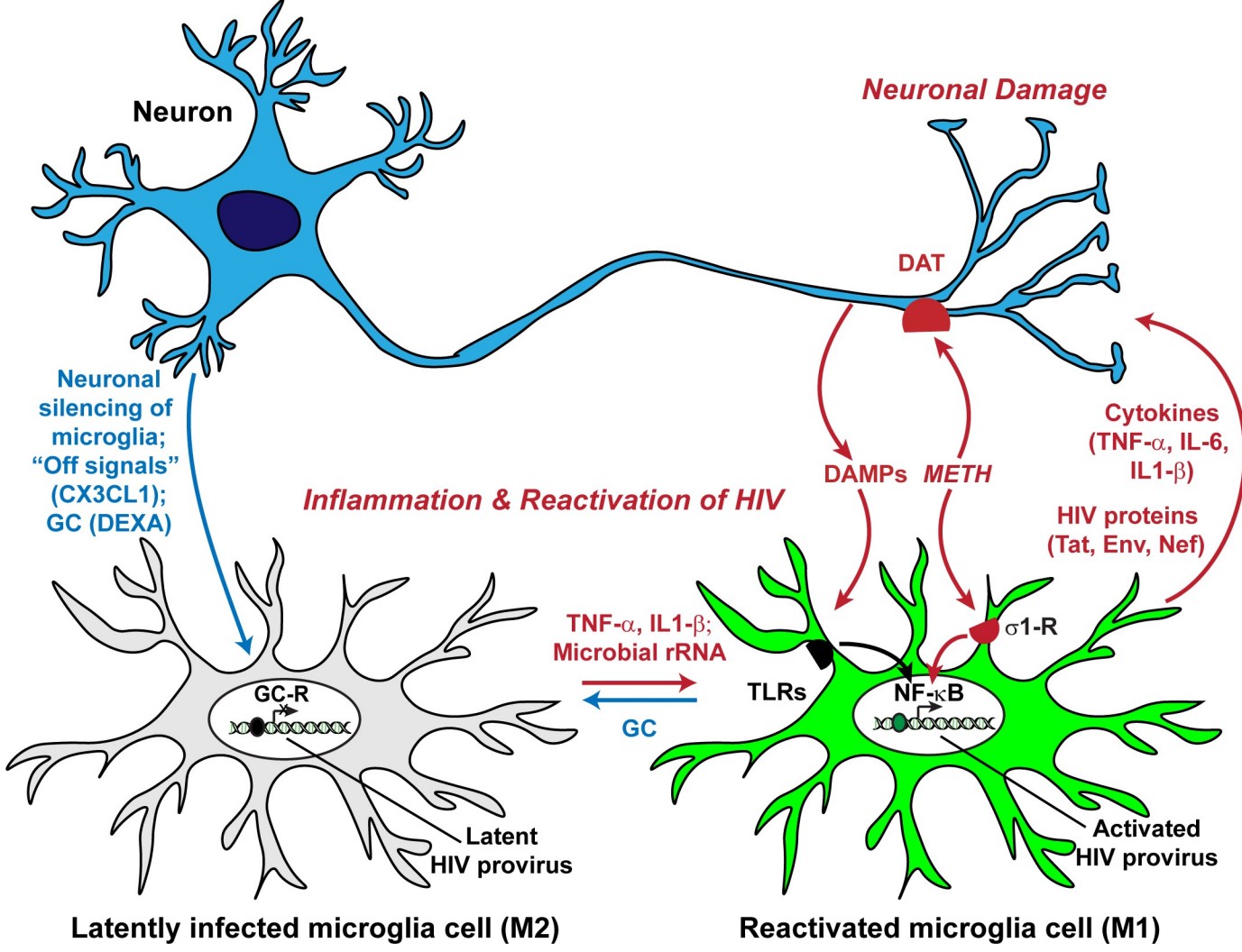

**Fig 12. Disruption of microglial cell-neuron communication drives HIV replication in the brain and leads to neuronal degeneration.** Healthy neurons suppress HIV expression in microglia. Although the mechanisms are not fully understood, HIV silencing correlates with establishment of a resting (M0) state. This is likely to be mediated by glucocorticoids (GC) and the fractalkine (CX3CL1) system. Inflammatory cytokines such as TNF-α and IL-1β, or TLR agonists such as LPS and microbial metabolites, including rRNA fragments, activate microglia (M1) and induce HIV transcription. The production of inflammatory cytokines and HIV proteins then leads to further neuronal damage. METH, acting through the σ1R receptor on microglia and DAT on neurons, works in concert with the pro-inflammatory agents to further disrupt normal cell physiology and enhance HIV transcription. Thus, inflammation can initiate a vicious cycle of neuronal damage/microglial cell activation leading to HIV reactivation.

## METH exacerbates neuronal damage and enhances HIV reactivation

METH dependence is one of the most common co-morbid conditions among the HIV-infected population [158], and several studies clearly demonstrate that HIV patients who abuse METH have exacerbated neurocognitive impairments [159, 160]. Acute use of METH can lead to psychological and behavioral abnormalities [161–164], while the deleterious effects of chronic METH use on neurons have been well documented in rodents, non-human primates [165–167], and humans [168]. Strong clinical evidence indicates that METH-induced immune activation in the CNS in combination with HIV effects increase neuronal injury in individuals with both risk factors [124, 125, 169, 170].

It is now widely recognized that microglia-mediated inflammation is critical in METH-induced neurotoxicity [171–174], and that microgliosis is the result of METH abuse that persists long after abstinence [175, 176]. Acute exposure of hμglia/HIV cells to high concentrations of METH does not induce immediate viral expression, as seen when latently infected cells are exposed to pro-inflammatory molecules such as TNF-α and poly (I:C). However, a significant level of HIV reactivation is observed after 96 h, suggesting that indirect autocrine mechanisms have been activated. Experiments performed with physiological concentrations of METH in drug abusers (for example at 100 nM; [122]) indicate that this low concentration is enough to induce HIV expression in microglia co-cultured with neurons as well as neuronal damage. Under this condition, inflammation, represented by activation of TLR3 on microglia, we found further increased HIV expression and exacerbation of neuronal damage.

We also demonstrated that METH acts through a specific receptor, σ1R, to reactivate HIV and trigger neurotoxicity in neuronal-microglial co-cultures. The selective σ1R antagonist BD1047 not only fully ameliorated METH-induced HIV reactivation, it also significantly reduced bystander neurotoxicity at 24 h. By contrast, the selective σ2R antagonist, SM-21 failed to restrict METH-induced increases in HIV expression. This finding indicates that METH acts through specific σ1R to reactivate HIV and trigger neurotoxicity, and strongly suggests σ1R is a key molecular site of METH and HIV comorbid interactions in neuroHIV. Since METH can act on both microglia and neurons through different receptors, a wide number of different molecular mechanisms can potentially play a role in regulating neuron-microglia communication in the context of METH abuse. Further experiments investigating these mechanisms are warranted, and they should include evaluation of the ON/OFF signaling through knock-out/knock-down strategies. Nonetheless, based on our findings and the work of others on the impact of METH on HIV replication [10, 14], it seems clear that chronic METH abuse enhances HIV-induced neuro-inflammation both through direct effects on σ1R-expressing microglia, and through indirect effects due to the induction of neuronal damage, leading to enhanced neurodegeneration, and an exacerbation of disease progression.

## Conclusions

Although systemic inflammation and antiretroviral and other drug toxicities are likely to contribute to the development of HAND, substantial evidence indicates that microglia, together with perivascular macrophages, are the main cellular reservoirs for HIV within the CNS parenchyma, and also contribute to HIV-related neuropathology and neurologic disorders. We have developed a reliable co-culture method to study the interactions between neurons and microglia and how inter-cellular signaling can regulate HIV latency in microglial cells. Our key observations are: First, that neurons can suppress viral expression, which likely involves repression of HIV through the GR. Second, that HIV-expressing infected microglia induce excessive neuronal damage, due to initiation of a vicious cycle of enhanced neuronal damage leading to enhanced microglial activation. Third, σ1Rs modulate the deleterious effects of METH on HIV reactivation and neurotoxicity. These results support the hypothesis that neurological complications of HAND result from the periodic emergence of HIV from latency within microglia in response to neuronal damage or inflammatory signals.

Bidirectional, interdependent cross-talk between neurons and microglia is therefore crucial in driving HIV infectivity in the CNS, and we believe that interrogating these signals is needed to identify basic mechanisms underlying the development of HAND. The recognition that HIV expression disrupts the normal interplay between microglia and neurons (i.e. the M0 to M1 transition cycle) and thereby exacerbates neurodegeneration should lead to the development of therapeutic anti-inflammatory strategies. Additionally, we believe that the screens of

receptor agonists and antagonists will help identify potentially novel therapeutic strategies to improve the clinical management of HAND, especially in HIV-infected individuals who abuse METH.

## Materials and methods

### Establishment of co-culture between LUHMES-derived neurons and hμglia/HIV

Primary human microglia are efficiently infected by replication competent R5 HIV (AD8gNef-GFP), and primary macaque microglia with replication competent SIV 17E-Fr particles [36]. However, due to the scarcity of primary microglia, for most of our studies here, we have use the HC69 clonal population of immortalized human microglial cells bearing an HIV construct (hμglia/HIV) [36, 37, 39] as a proxy for infected primary microglia. HC69 bears an HIV construct that lacks Gag, Pol and other accessory viral proteins. We have previously shown that the proviral integration site of the HC69 cells was sequenced and located within the host genome, demonstrating that HC69 is a single integrant and, as expected from the extensive studies characterizing HIV proviral integration sites, the provirus was located in the introns of host genes [39].

Undifferentiated, neuron precursor cells LUHMES/RFP, developed [100] and kindly provided by Dr. Stefan Schildknecht at the laboratory of Dr. Marcel Leist (Konstanz, Germany), contain a lentiviral construct bearing the sequence of red fluorescent protein (RFP) that allows visualization of neurite growth and its disturbance by toxicants. These cells were plated and allowed to expand for two days prior to transferring to the experimental wells at 500,000 neurons per well of a 6-well plate, or another quantity specified in the Figure Legends, for differentiation into dopaminergic neurons. One day after the neuronal differentiation process started, 60,000 immortalized human microglial cells bearing an HIV construct (Proviral HIV Structure) (hμglia/HIV; [36, 37, 39, 177]) were added. After 2 days, the classical dopaminergic markers are still in the induction phase, therefore LUHMES-derived neurons are still not considered full dopaminergic cells. After 3 days, these cells can then be considered maturing neurons, since they start producing classical markers of dopaminergic neurons, reaching their full mature status by day 6 [100, 178]. For the next 3 to 4 days, assays were performed to evaluate the viability of the neurons. All co-cultures with CHME-5 cells lasted only 48 h.

Where indicated, primary or iPSC-derived neurons were also used in co-culture with hμglia/HIV HC69, primary or iPSC-derived microglial cells. Primary human neurons (HN) and microglia were obtained from ScienCell Research Laboratories (CA), whereas human iPSC-derived microglia (Tempo-iMG), cortical (Tempo-iCort), dopaminergic (Tempo-iDopaNer), and motor (Tempo-iMotorNer) neurons were obtained from Tempo Bioscience (CA). These cells were plated, allowed to differentiate and maintained in culture on mouse laminin-coated plates or on Matrigel matrix (Corning), as directed by the manufacturer.

### Neuronal, microglia, and co-culture media

For expansion of undifferentiated, neuron precursors cells LUHMES/RFP [100], we used DMEM-F12 (ScienCell, Cat. #09411) supplemented with 1X N2 supplement (ThermoFisher Scientific, Cat. #17502048), 40 ng/mL bFGF (ThermoFisher Scientific, Cat. #13256029), 1% penicillin streptomycin (Gibco, Cat. #15070–063), and 250 ng/mL amphotericin B (Gibco, Cat. #15290018). For differentiation of LUHMES/RFP into dopaminergic neurons, DMEM-F12 was supplemented with 1 mM dibutyryl cAMP (StemCell Technologies, Cat. #73886), 1 μg/ml tetracycline (Sigma-Aldrich, Cat. #87128), 2 ng/mL GDNF (Sigma-Aldrich,

Cat. #SRP3309), 1% P/S, and 250 ng/mL amphotericin B. As recommended by the manufacturer, primary human neurons were grown in Neuronal Medium (NM, ScienCell Cat. #1521), primary human microglia in Microglia Medium (MM, ScienCell Cat. #1901), iPSC-derived microglia in DMEM/F-12 (LifeTech) supplemented with 1X N2 supplement (LifeTech), 0.5X NEAA (LifeTech), 2 mM L-Glutamine or 1X GlutaMax (LifeTech), 100 ng/mL GM-CSF (Peprotech), and 50 ng/mL IL-34 (Peprotech), iPSC-derived cortical neurons were grown in neurobasal-based medium supplemented with 1X B27 (LifeTech), 1X essential amino acids (LifeTech), 1X GlutaMax (LifeTech), and 10 μM all-trans-retinoic acid (Sigma Aldrich), iPSC-derived dopaminergic neurons also in neurobasal-based medium supplemented in 1X B27 (LifeTech), ascorbic acid (0.2 μM, Sigma Aldrich), cAMP (5 μM, Sigma Aldrich), and TGF-β3, recombinant human BDNF and GDNF (all at 10 ng/ml, Peprotech), and iPSC-derived motor neurons in DMEM/F12:neurobasal medium (50:50), supplemented with 1X B27 (LifeTech), 1X N2 (LifeTech), GlutaMax (0.5 mM (LifeTech), cAMP (5 μM, Sigma Aldrich), and recombinant human SHH, BDNF, GDNF, and CNTF (all at 10 ng/ml, Peprotech).

Unless otherwise noted, microglial cells (hμglia C20 and hμglia/HIV HC69; [36, 37, 39]) were cultured, as previously reported [39], in BrainPhys medium (StemCell Technologies, Cat. #05790) containing 1X N2 supplement, 1X penicillin streptomycin, 100 μg/mL normocin (InvivoGen, Cat. #ant-nr-1), 25 mM glutamine (Gibco, Cat. #25030081), 1% fetal bovine serum (FBS; Gibco, Cat. #10438026), and 1 μM DEXA (freshly added to the cell culture) (Sigma-Aldrich, Cat. #D4902). For the co-culture of microglia with neurons, the specific neuronal medium was used supplemented 1X insulin-transferrin-sodium selenite media supplement (Sigma-Aldrich, Cat. #I1884-1VL) and 0.2% FBS. Where indicated, METH, poly (I:C), alone or in combination were added to the HC69/primary HN co-cultures prior to collection of supernatant, and fixation for 15 min in 4% paraformaldehyde. MAP-2 was detected using rabbit anti-MAP-2 primary antibodies (1:500 dilution; Chemicon/EMD Millipore, Billerica, MA) followed by secondary goat anti-rabbit antibodies conjugated to Alexa Fluor-594 (red). Table 2 summarizes the composition of these formulations.

The reference for the culture of the other cell lines used as control, 293T cells, primary human foreskin fibroblasts, THP-1/HIV (A3) cells, and Jurkat/HIV (2D10) cells, are given in the corresponding figure legend.

To obtain damaged neuronal cells, cultured LUHMES-derived neurons were treated with 0.05% trypsin, followed by 1–3 min vortex, and cell death was verified by propidium iodide staining, as previously described [36]; at least 90% of cells were positive.

### Chemicals & reagents

To drive HIV emergence from latency and/or HIV expression increase, we used TNF-α (Invitrogen, Cat. #PHC3015), poly(I:C) (InvivoGen, #tlrl-pic), and METH (Sigma-Aldrich, Cat. #51-57-0) alone or in combination at the concentrations stated in the corresponding Figures and/or Figure Legends. To study the involvement of the σ receptors in METH-mediated HIV reactivation, BD1047 (Tocris, Cat. #0956), rimcazole (Tocris, Cat. #1497), and SM21 (Tocris, Cat. #0751) were used at the concentrations specified in the Figure Legends.

### SDS-PAGE/Western blot analysis

For determining the total expression of AMPA and phosphorylation of MAP2 and synapsin in LUHMES/RFP-derived neurons untreated or treated with TNF-α (10 pg/mL) or poly(I:C) (10 ng/mL) unexposed or not to C20 or HC69 cells untreated or treated with similar amount of TNF-α or poly (I:C), whole cell extracts (WCE) were prepared from $5 \times 10^5$ neurons collected by gentle shaking and added to RIPA buffer (25 mM Tris, pH 7–8, 150 mM Na, 0.1% SDS,

**Table 2. Medium composition for the growth and maintenance of LUHMES and microglial cells, and the co-culture between LUHMES-derived neurons and microglia.**

| Cells | Microglia or Neuronal Growth Medium (NGM) | Neuronal Differentiation Medium (NDM) | Modified Neuronal Differentiation Medium (mNDM) or Co-culture Medium |
|---|---|---|---|
| **Immortalized Microglia** | BrainPhys (w/o phenol) 1x N2 1x P/S 100 μg/mL normocin 2.5 mM glutamine 0.2%x FBS 1 μM dexamethasone | None | None |
| **Primary microglia** | Microglia Medium (MM, ScienCell Cat. #1901) | None | None |
| **iPSC-derived microglia** | DMEM/F-12 (LifeTech) 1X N2 supplement 0.5X NEAA 2 mM L-Glutamine 100 ng/mL GM-CSF 50 ng/mL IL-34 | None | None |
| **LUHMES** | DMEM-F12 1% N2 supplement 40 ng/mL bFGF 1% Pen/Strep 250 ng/mL amphotericin B | DMEM-F12 1 mM dibutyryl cAMP 1 μg/mL tetracycline 2 ng/mL GDNF 1% P/S 250 ng/mL amphotericin B | None |
| **Primary human neurons** | None | Neuronal Medium (NM, ScienCell Cat. #1521) | |
| **iPSC-derived cortical neurons** | None | Neurobasal 1X B27 1X non-essential amino acids 1X GlutaMax 10 μM all-trans-retinoic acid | None |
| **iPSC-derived dopaminergic neurons** | Neurobasal Medium 1X B27 0.2 μM ascorbic acid 5 μM cAMP 10 ng/mL TGFbeta3 10 ng/mL rhBDNF 10 ng/mL rhGDNF 10 μM cytosine arabinofuranoside | Neurobasal Medium 1X B27 0.2 μM ascorbic acid 5 μM cAMP 10 ng/mL TGFbeta3 10 ng/mL rhBDNF, 10 ng/mL rhGDNF | None |
| **iPSC-derived motor neurons** | None | 50:50 DMEM/F12: Neurobasal Medium 1X B27 1X N2 0.5 mM GlutaMax 5 μM cAMP 10 ng/mL rhSHH 10 ng/mL rhBDNF 10 ng/mL rhGDNF 10 ng/mL rhCNTF | None |
| **Immortalized microglia-LUHMES-derived neuronal co-culture** | None | None | NMD Insulin-transferrin-sodium selenite (Sigma-Aldrich I1884-1VL) 0.2% FBS |
| **Primary or iPSC-derived microglia-primary or iPSC-derived neurons** | None | None | 1:1 microglia medium/neuronal medium |

0.5% sodium deoxycholate, 1% Triton X-100). Protein concentration in WCE was measured by Bradford assay, and protein solutions were subjected to SDS-PAGE/Western blot using anti-glutamate receptor 1 (AMPA subtype) antibody [EPR5479] (Abcam, Cat. #ab109450), anti-MAP2 (phospho S136) antibody [EPR2361] (Abcam, Cat. #ab96378), and anti-synapsin I (Abcam, Cat. #phospho S9) antibody (Abcam, Cat. #ab194778). β-tubulin III (beta-TUJ) is a microtubule element of the tubulin family found exclusively in neurons [101, 102], therefore, we used anti-beta-TUJ (Abcam, Cat. #ab18207) as loading control. These primary antibodies were bound by the appropriate IRDye 800CW secondary antibody, and the membranes were scanned and analyzed using the Odyssey Infrared Imaging System (LI-COR Biosciences, NE).

### Flow cytometry and microscopy

As previously [36, 37, 39, 177], quantitation of GFP-expressing cells was carried out by fluorescence-activated cell sorting (FACS or flow cytometry) analysis using the LSRFortessa instrument for cell sorting, the FACSDiva software (BD, NJ) for data collection, and the WinList 3D software for data analysis, gating neurons separated from the microglia. To confirm microglia identity, we used anti-CD14-PE conjugated antibody (eBioscience, Cat. #12–0149), since we have shown that the CD14 receptor, a marker of monocytes, macrophages and dendritic cells [179], is present in the totality of the hμglia cells [36].

For phase contrast and fluorescence microscopy, we used the services of CWRU Visual Sciences Research Center core (grant #p30-15411373). In brief, a Leica DMI6000 Widefield Microscope with a QImaging EXI Aqua camera was used with a 1000 ms exposure time for FITC (488) to capture GFP fluorescence from HIV-expressing hμglia/HIV cells, and a 100 ms exposure time for the TxRed (546) to capture RFP fluorescence from neurons.

For neuronal cells immunostaining, we used anti-beta-TUJ (D71G9), XP rabbit (Cell Signaling Technology, #5568), anti-MAP2 rabbit (Cell Signaling Technology, #4542), anti-CXCR3 rabbit (Novus Biologicals, #NB10056404), anti-CD11b/c (Novus Biologicals, #NB11040766), anti-GAD 65/67 rabbit (Millipore, #AB1511), anti-dopamine transporter (DAT) (Millipore, #MAB369), and anti-acetylcholinesterase monoclonal (AchE) (Thermo, #MA3-042) antibodies. As secondary antibodies, Alexa Fluor 488 anti-rabbit or 594 anti-mouse were used.

### Neuronal viability assay

For viability of neuronal cells, we performed the *in-vitro* toxicology assay, based on resazurin reduction, following the instructions of the supplier (Sigma-Aldrich, Cat. #TOX8). Briefly, resazurin dye was added at a final concentration of 10% (v/v) to each well and incubated with the cells for ~2 hours, before transferring 100 μL of each well's supernatant to individual wells of a 96-well flat-bottom plate, and measuring absorbance (endpoint) at 600 nm and 690 nm (final absorbance = absorbance at 600- absorbance at 690). Healthy cells are able to reduce the dark blue resazurin dye, turning the solution to a lighter pink color. The lighter the dye the healthier and more viable the cells are, since they undergo more metabolic activity, which reduces the dye to a lighter pink color and, therefore, absorbance is lower. Quantitation was then calculated in percent using untreated neurons as reference. We also used healthy cell counting per field as endpoint for cell neuronal viability.

### Supporting information

**S1 Fig. Optimal co-culture medium.** LUHMES-derived neurons (red) and microglial cells, C20 (blue) and HC69 (green), were independently cultured in the presence of the indicated medium formulations (X-axis) and cell viability (Y-axis) was measured by the resazurin

method.
(TIF)

**S2 Fig. Microglia growth and viability is not affected in co-culture with neurons in a short-term.** (**A**) Growth rate. 60,000 hμglia/HIV HC69 cells were plated in the presence of increasing density of LUMHES-derived neurons (X-axis). After 24 h (short-term), neurons were killed with 0.25% trypsin for 30 seconds, and washed away with PBS prior to further trypsinization for 5 minutes to recover microglial cells. Cells were counted (Y-axis). (**B**) PI exclusion assay for measuring viability (Y-axis; right panel). N.S.: not significant.
(TIF)

**S3 Fig. Flow cytometry gating strategy for measuring CD14- and GFP-expressing hμglia/ HIV cells.** Flow cytometry profiles representing single cultures. The distinct populations of HC69 (μglia) and neuronal cells are indicated on the far-left flow cytometry profiles. Top flow cytometry profiles represent cells bound to isotype control; bottom profiles represent cells bound to anti-CD14 antibody. Anti-CD14 bound population is shown on the Y-axis, and GFP-expressing cells are shown on the X-axis in the CD14 vs. GFP graphs. The population of CD14-expressing cells is shown in orange and the populations of GFP-expressing cells are shown in green.
(TIF)

**S4 Fig. beta-TUJ immunochemistry.** LUHMES- and iPSC-derived neurons were stained with antibody against beta-TUJ (green). Alexa Fluor 488 anti-rabbit was used as secondary antibody. DAPI (blue) indicates nuclear staining.
(TIF)

**S5 Fig. MAP2 immunochemistry.** LUHMES- and iPSC-derived neurons were stained with antibody against MAP2 (green). Alexa Fluor 488 anti-rabbit was used as secondary antibody. DAPI (blue) indicates nuclear staining.
(TIF)

**S6 Fig. CXCR3 immunochemistry.** LUHMES- and iPSC-derived neurons were stained with antibody against CXCR3 (green). Alexa Fluor 488 anti-rabbit was used as secondary antibody. DAPI (blue) indicates nuclear staining.
(TIF)

**S7 Fig. CD11b/c immunochemistry.** LUHMES- and iPSC-derived neurons were stained with antibody against CD11b/c. Alexa Fluor 488 anti-rabbit was used as secondary antibody. DAPI (blue) indicates nuclear staining.
(TIF)

**S8 Fig. GAD65/67 immunochemistry.** LUHMES- and iPSC-derived neurons were stained with antibody against GAD65/67 (green). Alexa Fluor 488 anti-rabbit was used as secondary antibody. DAPI (blue) indicates nuclear staining.
(TIF)

**S9 Fig. DAT immunochemistry.** LUHMES- and iPSC-derived neurons were stained with antibody against DAT (green). Alexa Fluor 488 anti-rabbit was used as secondary antibody. DAPI (blue) indicates nuclear staining.
(TIF)

**S10 Fig. AchE immunochemistry.** LUHMES- and iPSC-derived neurons were stained with antibody against AchE (red or green). Alexa Fluor 488 anti-rabbit (green) or Alexa Fluor 594

anti-mouse was used as secondary antibodies. DAPI (blue) indicates nuclear staining.
(TIF)

**S11 Fig. 293T cells and human foreskin fibroblasts failed to induce HIV latency in HC69 cells.** (**A**) 60,000 hμglia/HIV HC69 cells were plated in the absence or presence of 0.5 x 10⁶ 293T cells or human foreskin fibroblasts (HFF), both grown in DMEM/10% FBS, or DEXA (positive control). The co-culture medium was the immortalized microglia medium (Table 2). HIV expression was evaluated after 24 h by flow cytometry. Flow cytometry profiles representing single cultures indicate the proportion of the CD11b/c-expressing cells (total microglia; Y-axis) that expresses GFP (X-axis). (**B**). Flow cytometric analysis quantification of microglial cell GFP expression in three similar experiments. The *p*-values of pair-sample, Student's *t*-tests comparing the microglial cells cultured alone or in the presence of cells are shown. Individual independent experiments are color coded (*n* = number of independent samples). N.S.: non-*s*ignificant. (**C**) PI exclusion assay to evaluate co-culture viability. Viability values (Y-axis) were normalized to the control culture of HC69 cells alone. Each colored symbol represents one experiment.
(TIF)

**S12 Fig. Neurons failed to induce HIV latency in THP-1/HIV and Jurkat/HIV cells.** 60,000 THP-1/HIV (A3) and Jurkat/HIV (2D10) cells, grown as previously described [37], were plated in the absence or presence of 0.5 x 10⁶ iCort or LUHMES-derived neurons, or U0126 (positive control). NDM (Table 2) was co-culture medium. HIV expression was evaluated after 24 h by fluorescence microscopy (**A**) and flow cytometry. (**B**). Flow cytometric analysis of GFP expression in three similar experiments: the *p*-values of pair-sample, Student's *t*-tests comparing the A3 and 2D10 cells cultured alone or in the presence of neurons are shown. Individual independent experiments are color-coded (*n* = number of independent samples). N.S.: non-*s*ignificant.
(TIF)

**S13 Fig. Microglia growth and viability is not affected in co-culture with neurons.** (**A**) Growth rate. 60,000 hμglia/HIV HC69 cells were plated in the presence of 0.5 x 10⁶ LUMHES-derived neurons (X-axis). After either 24 h or 72 h, neurons were killed with 0.25% trypsin for 30 seconds, and washed away with PBS prior to further trypsinization for 5 minutes to recover microglial cells. Cells were counted (Y-axis; left panel). (**B**) PI exclusion assay for measuring viability (Y-axis; right panel). N.S.: not significant.
(TIF)

**S14 Fig. Quantitation of the effect of healthy neurons vs. damaged neurons on HIV expression.** (**A**) hμglia/HIV HC69 cells were sorted into GFP⁻ cells. The population was expanded for 48 h prior to collection and co-cultured with either healthy neurons or damaged neurons (X-axis) at a ratio of 50:6. (**B**) GFP⁺ cells. Quantitation of GFP expression (Y-axis). Diamonds of similar color represent an individual experimental series. (*n* = number of individual samples). The *p*-values of pair-sample *t*-tests comparing the unexposed vs. the exposed cells are shown. N.S.: non-*s*ignificant.
(TIF)

**S15 Fig. HIV induces neuronal damage signals. A.** Western blot analysis. LUHMES-derived neurons were unexposed (-Microglia) or exposed to either C20 or HC69 cells in the absence or presence of either TNF-α or poly(I:C) for 48 h. Whole cell extracts were prepared from neurons and subjected to SDS-PAGE/Western blot analysis. Western blot membranes were blotted against anti-AMPA, anti-p-MAP2, and anti-p-synapsin antibodies, using anti-β-tubulin III

as loading control. Approximate molecular weights are indicated in KDa. **B.** Quantitation of AMPA, p-MAP2, and p-synapsin band intensity. Numbers were plotted in Relative Intensity (Arbitrary Units; Y-axis) vs. experimental treatments (represented by a specific color; X-axis), graphs for each AMPA, p-MAP2, and p-synapsin. **C.** Resazurin reduction assay. The resazurin reduction values (Y-axis) plotted are referenced to the control culture (neurons only), set at 100%, next to the other experimental treatments (X-axis). For (**B**) and (**C**), the p-values of statistically significant pair-sample *t*-tests (at the 0.05 confidence level, where the difference of the sample means is significantly different from the test difference of zero) of three experiments ($n = 3$) comparing the neurons exposed to C20 and either TNF-α or poly(I:C) with the neurons exposed to HC69 and either TNF-α or poly (I:C), respectively. N.S. stands for non-significant. Similar geometric figures represent a unique experiment.
(TIF)

**S16 Fig. Activated rat CHME-5/HIV cells exacerbate neuronal damage. A.** LUHMES-derived neurons were co-cultured or not with CHME-5 or CHME-5/HIV cells for 48 h. TNF-α (100 ng/mL) was added or not, as indicated. The neurons or co-cultures were stained with anti-MAP2 antibody followed by anti-rabbit Alexa Fluor 594 antibody (red) and DAPI for nuclear visualization. Green (GFP) depicts activated CHME-5/HIV cells. Dendrites and microglia are indicated by the white lines. **B.** LUHMES-derived neurons were co-cultured with CHME-5 (blue squares) or CHME-5/HIV (red triangles) cells that had been pre-activated with 100 ng/mL of TNF-α. Neurons alone (black circles) were used as control. Time-lapse images were taken every four hours, at the indicated time points, from 0 to 48 h (X-axis). The number of healthy neurons was counted in every field the relative number of viable neurons quantified (Y-axis).
(TIF)

**S17 Fig. METH-mediated reactivation of HIV.** Flow cytometry profiles representing single cultures of HC69 cells were incubated for 24, 48 or 96 h. (**A**) Untreated. (**B**) 300 μM METH. (**C**) 100 pg/ml TNF-α. (**D**) 100 ng/mL poly (I:C). GFP$^+$ cell populations were measured by flow cytometry and indicated in *bright green*.
(TIF)

**S18 Fig. METH sensitizes hμglia for poly (I:C)-mediated HIV reactivation.** HC69 cells were either untreated (- METH) or treated with METH 300 μM for 72 h prior to exposure to either TNF-α 20 pg/mL) or poly(I:C) (50 ng/mL) for another 24 h. Diamonds of similar color represent an individual experimental series. ($n$ = number of individual samples). The *p*-values of pair-sample t-tests comparing the unexposed vs. the exposed cells are shown. N.S.: non-significant.
(TIF)

**S19 Fig. HIV exacerbates METH-mediated neuronal damage.** Human neuronal and glial mixed-cultures containing astrocytes (Advanced Biosci. Resources) were maintained for 17 days *in vitro* (DIV) in BrainPhys supplemented with insulin-transferrin-sodium selenite prior to co-culture with either C20 or HC69 cells in either the absence or presence of 300 μM METH for 72 h. Top: brightfield. Middle: Green fluorescence channel. Bottom: Green (GFP$^+$ cells). Red (MAP2, neuronal dendrites). Blue (DAPI, all nuclei).
(TIF)

**S20 Fig. Effect of METH, TNF-α and poly(I:C) on neuronal damage.** LUHMES-derived neurons were either cultured alone (red) or co-cultured with either C20 (blue) or HC69 (green) cells in either the absence (control) or presence of (**A**) TNF-α, poly (I:C), METH,

METH + TNF-α or METH + poly (I:C), or (**B**) METH, BD1047, METH + BD1047 or DEXA for 72 h (X-axis) prior to neuronal survival quantitation by the resazurin method (Y-axis). $MPP^+$ was used as positive control for neuronal damage.
(TIF)

## Acknowledgments

We thank all past and present members of the Karn, Knapp, and Hauser laboratories for their feedback, suggestions, and useful discussions. We also thank the CWRU SOM Light Microscopy Imaging Core for their input in producing the fluorescence images.

## Author Contributions

**Conceptualization:** David Alvarez-Carbonell, Yoelvis Garcia-Mesa, Pamela E. Knapp, Kurt F. Hauser, Jonathan Karn.

**Data curation:** David Alvarez-Carbonell, Fengchun Ye, Pamela E. Knapp, Jonathan Karn.

**Formal analysis:** David Alvarez-Carbonell, Nirmala Ramanath, Pamela E. Knapp, Kurt F. Hauser, Jonathan Karn.

**Funding acquisition:** Kurt F. Hauser, Jonathan Karn.

**Investigation:** David Alvarez-Carbonell, Fengchun Ye, Nirmala Ramanath, Yoelvis Garcia-Mesa, Pamela E. Knapp, Kurt F. Hauser, Jonathan Karn.

**Methodology:** David Alvarez-Carbonell, Fengchun Ye, Nirmala Ramanath, Yoelvis Garcia-Mesa, Pamela E. Knapp, Kurt F. Hauser, Jonathan Karn.

**Project administration:** Pamela E. Knapp, Kurt F. Hauser.

**Resources:** Kurt F. Hauser, Jonathan Karn.

**Supervision:** Jonathan Karn.

**Validation:** Pamela E. Knapp, Kurt F. Hauser, Jonathan Karn.

**Writing – original draft:** David Alvarez-Carbonell, Jonathan Karn.

**Writing – review & editing:** David Alvarez-Carbonell, Fengchun Ye, Yoelvis Garcia-Mesa, Pamela E. Knapp, Kurt F. Hauser.

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
