## [Decision Letter · Decision Letter 0]

7 Jul 2019

Dear Prof. Karn,

Thank you very much for submitting your manuscript "Cross-talk between microglia and neurons regulates HIV latency" (PPATHOGENS-D-19-00977) for review by PLOS Pathogens. Your manuscript was fully evaluated at the editorial level and by independent peer reviewers. The reviewers appreciated the attention to an important problem, but raised some substantial concerns about the manuscript as it currently stands. These issues must be addressed before we would be willing to consider a revised version of your study. We cannot, of course, promise publication at that time.

We therefore ask you to modify the manuscript according to the reviewers' recommendations. Your revisions should address the specific points made by each reviewer; given the importance of the issues raised it will be necessary to send your manuscript back out for review.  Under such circumstances it will be important for the reviewers to feel the paper is very close to "publication-ready" with the second submission to consider acceptance.  One reviewer raised concerns about the form of the manuscript (with its 170+ references) which seems like a distraction to the science.  More importantly, the reviewers asked in a number of different ways to enhance our level of understanding of relevance of your model system.  This question is at the crux of how useful the observations are so it will be important to revise the manuscript (including new experiments) to address the concerns of the reviewers.

(1) A letter containing a detailed list of your responses to the review comments and a description of the changes you have made in the manuscript. Please note while forming your response, if your article is accepted, you may have the opportunity to make the peer review history publicly available. The record will include editor decision letters (with reviews) and your responses to reviewer comments. If eligible, we will contact you to opt in or out.

(2) Two versions of the manuscript: one with either highlights or tracked changes denoting where the text has been changed; the other a clean version (uploaded as the manuscript file).

Additionally, to enhance the reproducibility of your results, PLOS recommends that you deposit your laboratory protocols in protocols.io, where a protocol can be assigned its own identifier (DOI) such that it can be cited independently in the future. For instructions see http://journals.plos.org/plospathogens/s/submission-guidelines#loc-materials-and-methods

We hope to receive your revised manuscript within 60 days. If you anticipate any delay in its return, we ask that you let us know the expected resubmission date by replying to this email. Revised manuscripts received beyond 60 days may require evaluation and peer review similar to that applied to newly submitted manuscripts.

[LINK]

Sincerely,

Ronald Swanstrom

Associate Editor

PLOS Pathogens

Thomas Hope

Section Editor

PLOS Pathogens

Kasturi Haldar

Editor-in-Chief

PLOS Pathogens

orcid.org/0000-0001-5065-158X

Grant McFadden

Editor-in-Chief

PLOS Pathogens

orcid.org/0000-0002-2556-3526

Reviewer's Responses to Questions

**Part I - Summary**

Reviewer #1: This is a very interesting manuscript that addresses an important and under researched topic. The findings are of considerable interest and raise important questions regarding the ability of neurons to exert control over HIV replication in the nervous system. The use of a novel microglial HIV reporter line is a strength that facilitates the ability to perform the experiments. However, the use of cell lines and a model expression system is also a limitation as it may not accurately reflect HIV production and release. The general design of the studies and the rationale are strong but are impacted by a number of weaknesses. Neural damage can influence microglial replication but growth and death of microglia is not addressed. Overall, the manuscript is not well focused and lacks clarity in many sections as outlined below.

Reviewer #2: The manuscript by Alvarez-Carbonell examines whether the cross talk between microglia and neurons using a co-culture model system, where a microglia cell line HC69 that has a HIV reporter system were co-cultured with different types of neurons including LUHMES/RFP neurons, differentiated cortical, dopaminergic and motor neurons as well as mouse primary neurons. The authors analyzed whether co-culture of neurons with HC69 cells altered the GFP expression in these cells under different conditions and in combination with METH. The author conclude that neurons induce HIV latency, but that dying neurons then reverses this phenotype. The question the authors are addressing is important, however, the authors only use one system (GFP expression of the lentiviral reporter) to address this and do not validate their studies with any other system

Reviewer #3: There is a great deal of interest is examining viral latency in the CNS, however, there are few tools for doing so. Alvarez-Carbonell et al have developed an in vitro system to examine how interactions between neurons and microglia influence HIV expression and neuronal damage. This system involves their previously developed immortalized microglia cell line that has been infected with a VSV-g pseudotyped reporter vector. This cell line now harbors a partial HIV genome and GFP under control of the HIV LTR. In this study they use this immortalized microglia cell line and neuronal cells lines to examine whether interactions between these cells influences viral latency. Based on experiments co-culturing these cells they observed that HIV expression in microglia is lower when healthy neurons are present, but HIV expression increases when damaged neurons are present. Further, their results suggest that neuron types differ in their ability to suppress HIV expression and that other substances can stimulate HIV expression including TNF-a and meth. Together, these results suggest that HIV latency in the CNS may be highly regulated by both local cells that suppress expression when healthy, increase expression when damaged and increased expression when the environment becomes inflammatory. These results suggest that CNS reservoirs may be ‘shocked’ by unique triggers that may not alter latency in CD4+ T cells.

Reviewer #4: - strong/credible previous work from this group establishing their model of immortalized brain microglial cells

- co-culture approach is a vaulable extension to the microglia-only work and, while a cell culture model with the limitations of that simplified system, allows for a window into interactions between microglia and neurons that has value

- co-culture of neurons and microglia has been standardly used in other models

-finding that damaged neurons slightly increased HIV expression in the microglia is interesting and plausible

-overall the finding that the condition of the neurons impacts HIV expression is novel and provocative, though there are some limitations in interpreting these data

**Part II – Major Issues: Key Experiments Required for Acceptance**

Reviewer #1: Neuron viability is assessed but not microglial which is more important for interpretation of the results. The authors need to address the question of microglial growth/survival. The growth rate of the immortalized microglia may be sufficient to have an impact on the interpretation of the studies. Any differential growth or death could alter the % of GFP expressing cells.

Meth concentrations used in the study are very high - at least an order of magnitude higher than plasma concentrations in Meth abusers. Data from studies in the low µM range would provide more translatable conclusions.

Reviewer #2: 1) There is a lack of explanation of the system in the introduction or methods and the lack of a verification with any other type of system. The authors use GFP+ cells as the sole read-out of HIV latency. They also do not describe the system in this manuscript. In their previous paper (Garcia-Mesa et al. JNV 2017), the % of cells expressing GFP was very low (<5%). In this paper, the HC69 cells have varied expression of GFP in normal culture. Is this due to cell division (ie, do dividing cells have reduced GFP which then rebounds). The authors use GFP- and GFP+ sorted population and compare the differences between them without providing any information on what is regulating this GFP expression in the absence of any other stimuli.

2) Additionally, the “induced latency” is only at a single time point (24 hrs) of decreased # of cells expressing GFP to declare that neurons induce HIV latency. A more clear kinetics of GFP expression (6 hr, 12 hr, 24 hr, 36 hr) would provide better evidence of suppressed GFP production.

3) One of the strongest correlates with the loss of suppression is the loss of neuronal viability. Could it not simply be that the neurons are dying and therefore there are less neurons to suppress latency. The authors suggest strongly in the abstract that damage neurons increase HIV suppression, but they would have to add damaged neurons to a normal co-culture to show this.

4) The authors show a substantial drop in live neuron counts at 72 hrs in their co-culture system (Fig. 6). However, in the study to examine neuronal damage, there was little reduction in cell numbers (Fig. 7), with reduced dendrites being the main criteria for damaged neurons. This is a slightly different system, but a huge difference in outcomes that appears to be primarily glossed over. It should be discussed.

Reviewer #3: Studying viral reservoirs in the CNS is a complicated proposition that likely involves the use of suboptimal in vitro systems or human samples. My primary concern with this study is that the model in question has deviated so much from in vivo biology as to be uninformative. In this study the authors examine interactions between two cell lines rather than primary cells. In their previous work the authors found that expression in these cell lines differs substantially from that of primary cells. In addition, both cell lines are infected with lentiviral vectors. The microglia are infected with a partial HIV reporter genome (see below) and the neurons are infected with a lentivirus reporter vector that is not/poorly described. The authors clearly show that expression of a partial HIV genome changes in culture conditions that include neurons. However, these experiments often lack the necessary controls to assess the cause of these changes. Is it due to interactions that are specific to microglia and neurons? Or is it due to the high density of cells in some culture conditions (see below) or interactions between microglia and any cells type (also see below). Further, would the effects be the same if the HIV reporter virus was intact? These types of questions should be addressed in order to assess whether this model captures features of the in vivo biology. Without these controls it is very difficult to assess whether these provocative observations represent specific interactions that likely occur in the CNS in vivo or are a product of the culture conditions.

1. The methods are insufficiently described. I am assuming that this study infects cells using a VSV-g pseudotyped vector, as in their previous work (PMC5294768), but that needs to be clarified. Infection with a VSV-g pseudotyped variant would not represent a true HIV Env-mediated infection. Did the authors choose this approach because it generates a higher rate of infection than infection with reporters pseudotyped with an actual HIV Env? I have three primary concerns with it. a) It suggests that HIV may infect microglia very inefficiently; raising questions about the importance of this event. b) If a very high percentage of cells are infected, we would expect that some cells are infected with multiple virions and may express more viral proteins than cells with a single integrated provirus. c) The vector used in this study doesn’t encode all of the viral proteins. This is unfortunate given that the study is examining cellular interactions that may be mediated by viral proteins. All of these points should be discussed.

2. Did the authors examine whether the results outlined in Figure 2 are specific to microglia culture with neurons? If I understand correctly, as the ratio of neurons to microglia increased, the total number of cells also increased. Is it possible that the reduction in GFP expression when the ratio of neurons to microglia is high could be due to there being a larger number of cells in the coculture?

3. Similar to the point above, do the authors examine whether the effect observed in Figure 4 is specific to neurons or influenced by the total density of cells in the culture? There are many ways to address this. It would be nice to try a couple of the following:

• Keep the total cell density constant across treatments (I don’t think that the authors did this).

• Repeat experiment with cells other than neurons

• Culture in a trans-well system where the density of microglia is constant but factors produced by neurons can reach the microglia culture or add spent media from neuron cultures to the microglia.

4. The authors should confirm that the neurons also mediate HIV expression in primary microglia. In a previous study (PMC5329090) the authors observed that their immortalized human microglia have gene expression patterns that differ substantially from that of primary microglia.

5. The authors should discuss the significance of the five types of neurons examined in Figure 3. Were they are terminally differentiated?

6. The author argue that neurons block HIV expression. They should also examine whether this is due to changes in global expression.

7. It would be helpful if the authors listed samples sizes. For example, does each flow plot in figure 8 represent a single culture?

8. What does including figures 7 and 9 accomplish? Summary figures would be more informative. I suggest moving 7 and 9 to the supplemental materials.

9. Figure 10 is very hypothetical. This hypothesis provides easily tested predictions that glucocorticoids can influence HIV expression. In the absence of additional evidence, I recommend that Figure 10 be moved to the supplemental materials.

10. Lines 593-595: “Substantial evidence indicates that microglia, together with perivascular macrophages, are the main cellular reservoirs for HIV within the CNS parenchyma, and hence responsible for HIV-related neuropathology and neurologic disorders.” This is a very strong statement suggesting that HAND is due to infection of macrophage/microglia. I would recommend softening this statement given that this mechanism has not been demonstrated in vivo and the fact that other mechanisms such as drug toxicities haven’t been ruled-out.

Reviewer #4: -HIV 'silencing' by increasing concentrations of neurons:microglia is a surprising findings and can't help but raise more questions than it answers. Why cortical and dopaminergic neurons but not motor neurons woudl selectively reduce HIV expression is unexplained/unexplored

- short term silencing and 'long term' considered 72 hoursin this cell culture system has questionable significance for human disease

- Methamphetamine inclusion in this study is not well explained/justified. Further experiments investigating the mechanisms of the neuronal effects on microglia and HIV expression would be more relevant to the current paper. I would consider removing the methamphetamine experiments from the paper and focusing on the huglia/neuronal interactions.

**Part III – Minor Issues: Editorial and Data Presentation Modifications**

Reviewer #1: 1) Too much of the manuscript contains general information and reads like a review rather than a research paper. Along the same line, the reference list is exceptionally long. The authors should focus the manuscript by eliminating background material that is not directly relevant with a corresponding reduction in the number of references.

2) Resolution of the images in Fig 1 is not very good. Higher quality images would improve the ability to see what the cells actually look like.

3) The concentration of FBS needs to be clarified. It is listed as 0.1% 0.2% or 1% in different places.

4) The authors should define LUHMES derived neurons and add a justification for their use.

5) It would be useful to determine if the cellular interaction effects are specific to neurons. Addition of the microglia to other cells types would help to solidify the argument.

6) The role of neuronal damage in GFP expression is confusing. In Fig 4 there is substantial neuron death associated with decreased GFP expression whereas in Fig 5 & 6 neural damage/death is associated with increased GFP expression.

7) In Fig 7 it is difficult to tell what is circled. In addition, it appears that the HC69 cells were pre-activated to induce HIV expression. Under these circumstances it is unclear how the resulting toxicity can be attributed to HIV induction. A comparison of GFP+ vs GFP- HC69 cells without added cytokines would provide a cleaner comparison.

8) In the supplemental Figs, the cortical, dopamine and motor “neurons” do not look neuronal which raises questions about the adequacy of the differentiation. How do the authors know these are functional neurons? The addition of primary neurons to the experiments would add additional credibility.

9) It would be useful to know if the effect of the neurons on the microglia is contact dependent. Does neuronal conditioned medium have the same effect?

10) It is not clear why the paradigm shifted to use of CHME-5 cells. The authors should provide a rationale. Also in supplemental Fig 11, it appears that the neural damage is restricted to TNF treated cells not the HIV expressing cells. This again suggests that cytokine treatment and not HIV expression is the critical variable (which is already well established).

11) Are microglia added while neurons are in the differentiation medium? It is not clear when and how medium changes take place. (Neurobasal is misspelled in the Table)

12) Exposure to the different differentiation factors may confound the interpretation of the experiments. The culture conditions are very complex and it is not clear how the effects of growth factors and other components can be factored out.

13) Figure 10 is very speculative and not evidence based. The conclusions should be restricted to what is supported by data from the study.

14) How does the proposed involvement of glucocorticoids relate to this study?

15) Dexamethasone is in the microglial culture medium. Is this a confounding factor that needs to be addressed?

16) The conclusions that can be drawn are restricted by the nature of the model system. Addition of data on actual HIV production from infected cells would greatly strengthen the manuscript.

Reviewer #2: 1) The authors make statements of their system to indicate they are studying the effect of HIV infection (lines 104-105 “study the impact of HIV infection of microglia cells on their interaction with neurons” is just one example). Unless measurable virus is being produced and released, they are not studying the effect of virus infection on neurons. They are using a virus-reporter system, not virus production, which can have different effects on the cells. This needs to be put in context throughout the paper.

2) In Figure 2A, the GFP+ cells by themselves have cytoplasmic GFP expression, while cells cultured with neurons has what appears to be nuclear staining of GFP in the cells that are expressing it. However, the average GFP expression per positive cells seems similar (flow data). Does induction of latency affect GFP localization in the cell?

3) The authors should check with a statistician whether a One-way ANOVA would be the more correct statistical test for most of these studies.

4) Fig. 3C is not clear enough to show different cell populations

5) Box and whiskers are not needed on graphs when individual data points are given.

5) The discussion was basically a reiteration of the introduction until line 533.

Reviewer #3: Lines 467-470: “Pathophysiologically, HAND is thought to be driven the sustained, low-level production of HIV proteins and proinflammatory toxins that result in synaptodendritic injury, reductions in CNS connectivity with accompanying neurobehavioral and cognitive declines [123-126].” These are a couple of explanations, but may want to include others including drug toxicities (see PMC4937456).

Are the cells called μglia or mμglia?

Lines 59-62. There is an error in this sentence “In the US, 10-15% of HIV patients acknowledge METH use [8], exacerbates the effects of HIV infection in the CNS [9, 10] due to a combination of neurotoxic effects and the enhancement of HIV replication in microglia [11-14].”

Reviewer #4: (No Response)

PLOS authors have the option to publish the peer review history of their article (what does this mean?). If published, this will include your full peer review and any attached files.

Reviewer #1: No

Reviewer #2: No

Reviewer #3: No

Reviewer #4: No

---

## [Editor Report · Decision Letter 1]

1 Dec 2019

Dear Prof. Karn,

We are pleased to inform that your manuscript, "Cross-talk between microglia and neurons regulates HIV latency", has been editorially accepted for publication at PLOS Pathogens. 

Before your manuscript can be formally accepted and sent to production, you will need to complete our formatting changes, which you will receive by email within a week. Please note that your manuscript will not be scheduled for publication until you have made the required changes.

IMPORTANT NOTES

(1) Please note, once your paper is accepted, an uncorrected proof of your manuscript will be published online ahead of the final version, unless you’ve already opted out via the online submission form. If, for any reason, you do not want an earlier version of your manuscript published online or are unsure if you have already indicated as such, please let the journal staff know immediately at plospathogens@plos.org.

(2) Copyediting and Proofreading: The corresponding author will receive a typeset proof for review, to ensure errors have not been introduced during production. Please review the PDF proof of your manuscript carefully, as this is the last chance to correct any errors. Please note that major changes, or those which affect the scientific understanding of the work, will likely cause delays to the publication date of your manuscript. 

(3) Appropriate Figure Files: Please remove all name and figure # text from your figure files. Please also take this time to check that your figures are of high resolution, which will improve the readbility of your figures and help expedite your manuscript's publication. Please note that figures must have been originally created at 300dpi or higher. Do not manually increase the resolution of your files. For instructions on how to properly obtain high quality images, please review our Figure Guidelines, with examples at: http://journals.plos.org/plospathogens/s/figures.

(4) Striking Image: Please upload a striking still image to accompany your article if one is available (you can include a new image or an existing one from within your manuscript). Should your paper be accepted, this image will be considered for our monthly issue image and may also appear on our website to feature your article. Please upload this as a separate file, selecting "striking image" as the file type upon upload. Please also include a separate "Other" file with a caption, including credits and any potential copyright information. Please do not include the caption in the main article file. If your image is from someone other than yourself, please ensure that the artist has read and agreed to the terms and conditions of the Creative Commons Attribution License at http://journals.plos.org/plospathogens/s/content-license. Please note that PLOS cannot publish copyrighted images.

(5) Press Release or Related Media: If your institution or institutions have a press office, please notify them about your upcoming paper at this point, to enable them to help maximize its impact. If they will be preparing press materials for this manuscript, please inform our press team in advance at plospathogens@plos.org as soon as possible. We ask that you contact us within one week to plan ahead of our fast Production schedule. If you need to know your paper's publication date for related media purposes, you must coordinate with our press team, and your manuscript will remain under a strict press embargo until the publication date and time. This means an early version of your manuscript will not be published ahead of your final version. 

(6)  PLOS requires an ORCID iD for all corresponding authors on papers submitted after December 6th, 2016. Please ensure that you have an ORCID iD and that it is validated in Editorial Manager.  To do this, go to ‘Update my Information’ (in the upper left-hand corner of the main menu), and click on the Fetch/Validate link next to the ORCID field.  This will take you to the ORCID site and allow you to create a new iD or authenticate a pre-existing iD in Editorial Manager

(7) Update your Profile Information: Now that your manuscript has been provisionally accepted, please log into Editorial Manager and update your profile, if needed. Go to https://www.editorialmanager.com/ppathogens, log in, and click on the "Update My Information" link at the top of the page. Please update your user information to ensure an efficient production and billing process. 

(8) LaTeX users only: Our staff will ask you to upload a TEX file in addition to the PDF before the paper can be sent to typesetting, so please carefully review our Latex Guidelines http://journals.plos.org/plospathogens/s/latex in the meantime.

(9) If you have associated protocols in protocols.io, please ensure that you make them public before publication to guarantee immediate access to the methodological details.

Best regards,

Ronald Swanstrom

Associate Editor

PLOS Pathogens

Thomas Hope

Section Editor

PLOS Pathogens

Kasturi Haldar

Editor-in-Chief

PLOS Pathogens

orcid.org/0000-0001-5065-158X

Grant McFadden

Editor-in-Chief

PLOS Pathogens

orcid.org/0000-0002-2556-3526
---

## [Editor Report · Acceptance letter]

19 Dec 2019

Dear Prof. Karn,

We are delighted to inform you that your manuscript, "Cross-talk between microglia and neurons regulates HIV latency," has been formally accepted for publication in PLOS Pathogens.

Best regards,

Kasturi Haldar

Editor-in-Chief

PLOS Pathogens

orcid.org/0000-0001-5065-158X

Grant McFadden

Editor-in-Chief

PLOS Pathogens

orcid.org/0000-0002-2556-3526